# Fast-moving stars around an intermediate-mass black hole in ω Centauri

Maximilian Häberle[1 ✉], Nadine Neumayer[1], Anil Seth[2], Andrea Bellini[3], Mattia Libralato[4,5], Holger Baumgardt[6], Matthew Whitaker[2], Antoine Dumont[1], Mayte Alfaro-Cuello[7], Jay Anderson[3], Callie Clontz[1,2], Nikolay Kacharov[8], Sebastian Kamann[9], Anja Feldmeier-Krause[1,10], Antonino Milone[11], Maria Selina Nitschai[1], Renuka Pechetti[9] & Glenn van de Ven[10]

Black holes have been found over a wide range of masses, from stellar remnants with masses of 5–150 solar masses ($M_\odot$), to those found at the centres of galaxies with $M > 10^5 M_\odot$. However, only a few debated candidate black holes exist between $150 M_\odot$ and $10^5 M_\odot$. Determining the population of these intermediate-mass black holes is an important step towards understanding supermassive black hole formation in the early universe[1,2]. Several studies have claimed the detection of a central black hole in ω Centauri, the most massive globular cluster of the Milky Way[3–5]. However, these studies have been questioned because of the possible mass contribution of stellar mass black holes, their sensitivity to the cluster centre and the lack of fast-moving stars above the escape velocity[6–9]. Here we report the observations of seven fast-moving stars in the central 3 arcsec (0.08 pc) of ω Centauri. The velocities of the fast-moving stars are significantly higher than the expected central escape velocity of the star cluster, so their presence can be explained only by being bound to a massive black hole. From the velocities alone, we can infer a firm lower limit of the black hole mass of about $8{,}200 M_\odot$, making this a good case for an intermediate-mass black hole in the local universe.

ω Centauri (ω Cen) is a special case among the globular clusters of the Milky Way. Owing to its high mass, complex stellar populations and kinematics, ω Cen is widely accepted as the stripped nucleus of an accreted dwarf galaxy[10,11]. These factors combined with its proximity ($D = 5.43$ kiloparsec (kpc); ref. 12) have made it a prime target for searching for an intermediate-mass black hole (IMBH). As part of the oMEGACat project[13,14], we recently constructed an updated proper-motion catalogue of the inner regions of ω Cen, on the basis of more than 500 Hubble Space Telescope (HST) archival images taken over a time span of 20 years. The unprecedented depth and precision of this catalogue have enabled us to discover a statistically significant overdensity of fast-moving stars in the centre of the cluster (Fig. 1 and Extended Data Fig. 1). In total, we find seven stars with a total proper motion higher than 2.41 mas yr$^{-1}$ within 3″ of the centre determined in refs. 6,15 (hereafter AvdM10 centre). At a cluster distance of 5.43 kpc (ref. 12), this corresponds to projected two-dimensional (2D) velocities higher than the escape velocity of the cluster if no IMBH is present ($v_{esc} = 62$ km s$^{-1}$; ref. 16; Methods). Here we show that the presence of these stars strongly indicates a massive black hole, similar to the S-stars in the Galactic centre[17]. A list of the fast-moving stars is provided in Extended Data Table 1, and we label the fast-moving stars with letters from A to G, sorted by their proximity to the AvdM10 centre. All these stars lie along the cluster main sequence in the colour–magnitude diagram (CMD) (Fig. 2). The fastest and the centremost star (Fig. 1, star A) has a 2D proper motion of $4.41 \pm 0.08$ mas yr$^{-1}$ ($113.0 \pm 1.1$ km s$^{-1}$). The motion of this star was measured over 286 epochs and a full 20.6-year time baseline (Fig. 3). We run extensive quality checks to ensure that the astrometry of the discovered fast-moving stars is reliable. To ensure the cleanest possible dataset, we limit our analysis to stars whose velocity is at least $3\sigma$ above the escape velocity. This leads to the exclusion of stars B and G; however, this has negligible influence on the determined IMBH constraints.

Four of the fast-moving stars, including the three fastest in the sample, are found within the centremost arcsecond ($r_{projected} < 0.03$ pc or $r_{projected} < 0.09$ ly). These four innermost stars are all fainter than $m_{F606W} > 22.7$, which is unlikely ($P = 0.013$) to be a random occurrence given the overall distribution of stellar magnitudes in the centre of ω Cen. Moreover, all of them lie towards the blue side of the main sequence. Both these properties could have interesting physical implications on the mechanism involved in capturing these stars or on their tidal interactions with the IMBH.

We expect a certain number of Milky Way stars in our field of view, and because they have a large proper motion with respect to ω Cen, they can mimic fast-moving cluster stars. On the basis of the number

[1]Max Planck Institute for Astronomy, Heidelberg, Germany. [2]Department of Physics and Astronomy, University of Utah, Salt Lake City, UT, USA. [3]Space Telescope Science Institute, Baltimore, MD, USA. [4]AURA for the European Space Agency (ESA), Space Telescope Science Institute, Baltimore, MD, USA. [5]INAF, Osservatorio Astronomico di Padova, Padova, Italy. [6]School of Mathematics and Physics, The University of Queensland, St Lucia, Queensland, Australia. [7]Facultad de Ingeniería y Arquitectura, Universidad Central de Chile, La Serena, Chile. [8]Leibniz Institute for Astrophysics, Potsdam, Germany. [9]Astrophysics Research Institute, Liverpool John Moores University, Liverpool, United Kingdom. [10]Department of Astrophysics, University of Vienna, Wien, Austria. [11]Dipartimento di Fisica e Astronomia 'Galileo Galilei', Università Degli Studi di Padova, Padova, Italy. ✉e-mail: haeberle@mpia.de

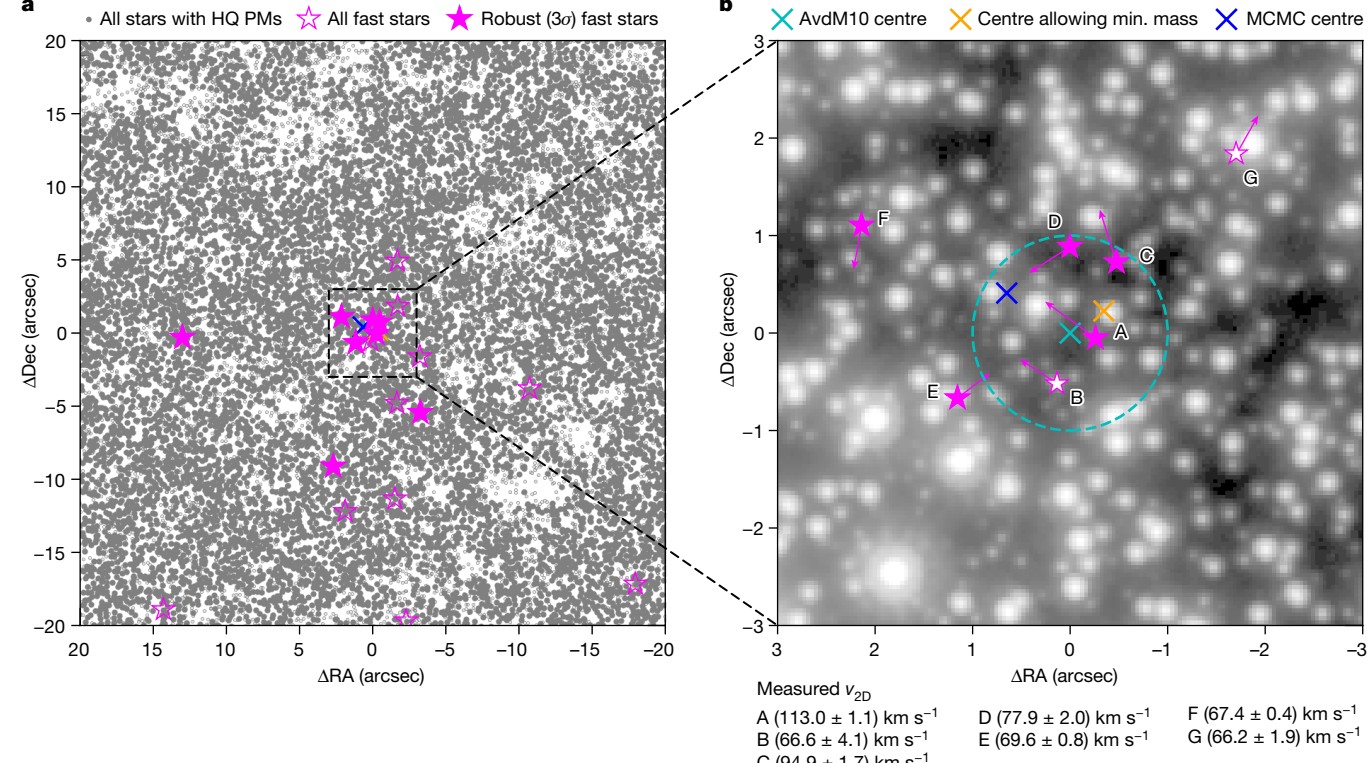

**Fig. 1 | Location of fast-moving stars. a**, Solid grey markers indicate all stars with a high-quality (HQ) proper motion measurement in our new catalogue within a 40″ × 40″ region centred on the AvdM10 centre[6]. Well-measured stars with velocities higher than the cluster escape velocity (62 km s⁻¹) and lying on the cluster main sequence in the colour-magnitude diagram (Fig. 2) are marked in pink. We use filled markers for stars that are at least 3σ above the escape velocity, and open markers for all other stars above the escape velocity. **b**, Stacked image of the innermost region of ω Cen using all observations in the WFC3/UVIS F606W filter. The fast-moving stars and their proper-motion

vectors are shown in pink. The arrows indicate the stellar motion over 100 years. We also list the measured individual velocities. The cyan cross indicates the photometric centre of ω Cen measured by ref. 6, with the dashed circle indicating the 1″ error reported for this centre; the orange cross marks the centre, allowing for the lowest IMBH mass; and the blue cross marks the most likely position of an IMBH given the Markov chain Monte Carlo (MCMC) analysis of the acceleration limits of the fast-moving stars. Dec, declination; RA, right ascension, min., minimum.

density of fast-moving stars at larger radii (Extended Data Fig. 1), we estimate the rate of contaminants to be 0.0026 arcsec⁻², which is consistent with the expectations from the Besançon Milky Way model[18]. This number density gives an expected average value of only 0.074 foreground stars in the inner 3-arcsec radius. A detection of five such stars by a pure coincidence can, therefore, strongly be ruled out by simple Poisson statistics ($P = 1.7 \times 10^{-8}$) (ref. 19). Having two or more random contaminants within our five-star sample can also be ruled out at the 3σ level ($P = 0.0026$). We also show in the Methods that these stars cannot be explained by objects bound to stellar mass ($\lesssim 100 M_\odot$) black holes and that ejections from three-body interactions or an IMBH are not plausible.

Therefore, the presence of the seven central stars moving faster than the escape velocity of the cluster can be explained only if they are bound to a compact massive object near the centre, raising the local escape velocity. If no massive object was present, their velocities would cause them to leave the central region in less than 1,000 years, and then eventually escape the cluster. These fast-moving stars are a predicted consequence of an IMBH, but these stars are not expected from mass-segregated stellar-mass black holes[9].

We do not know several parameters of the system, including the mass and exact location of this massive object, the relative line-of-sight (LOS) distance between the object and the stars, and the LOS velocity of the stars. Despite this, we can calculate a lower limit on the mass of the dark object using the 2D velocities of the fast-moving stars, assuming only that they are bound to it. The combined constraint from the velocities of the five robustly measured fast-moving stars is about

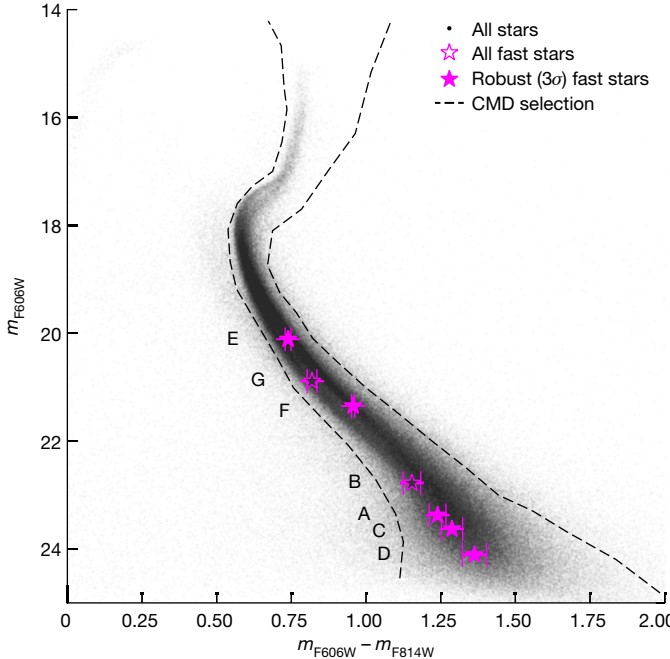

**Fig. 2 | HST-based colour-magnitude diagram (CMD) of ω Cen.** The CMD locations of all fast-moving stars are marked with pink symbols, with their photometric 1σ errors marked with error bars. All of them lie on the main sequence, showing that they are probably members of ω Cen. The stars are labelled from A to G, sorted by their distance from the AvdM10 centre.

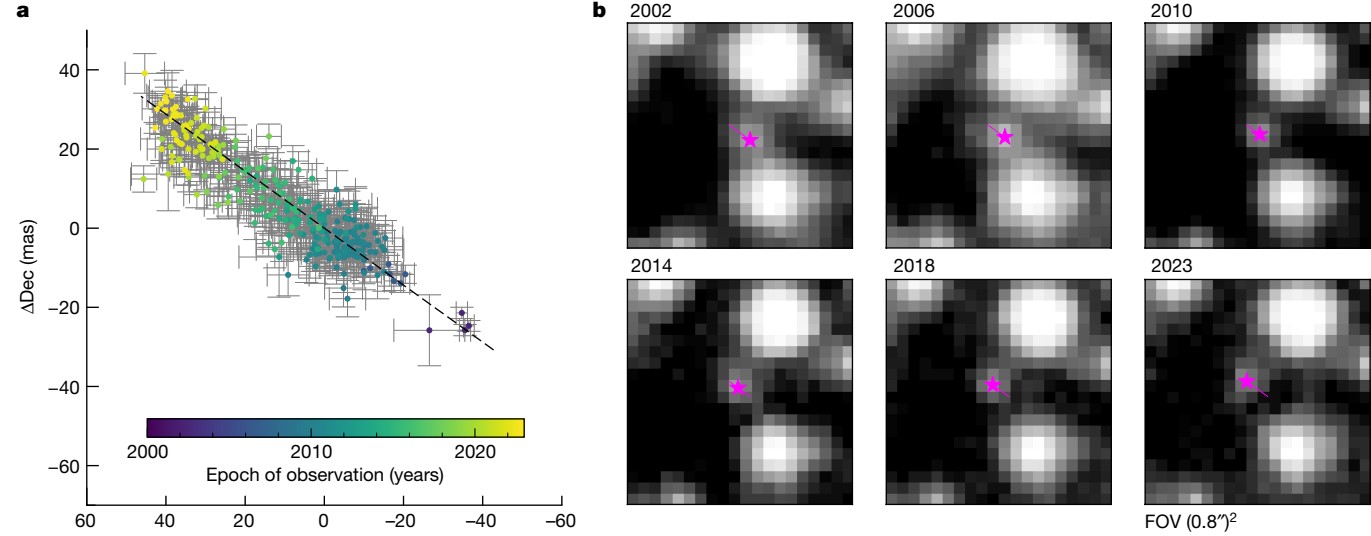

**a**

**b**

2002  2006  2010

2014  2018  2023

FOV (0.8″)²

**Fig. 3 | Motion of the fastest star. a**, Individual measured positions with 1$\sigma$ error bars. **b**, Multi-epoch HST imaging for star A, the fastest ($v_{projected}$ = 113.0 ± 1.1 km s⁻¹) and centremost of the seven fast-moving stars discovered in the centre of ω Cen. The star is indicated with a pink marker on the images and its motion over 21 years with a line. This plot shows the notable astrometric quality and the long temporal baseline we have in our unique dataset. Similar plots for all other stars are shown in Extended Data Fig. 2. Dec, declination; RA, right ascension; FOV, field of view.

8,200$M_\odot$, thus making an IMBH the only plausible solution. The position of the IMBH requiring the lowest mass is only 0.3 arcsec away from the AvdM10 (ref. 6) centre, in agreement with the ±1″ error on the AvdM10 centre. Further details of this calculation can be found in the Methods.

Although the linear motion and the velocity of the stars can be measured with great precision, the expected acceleration signal from an IMBH is considerably weaker and harder to detect. However, even a non-detection of acceleration could provide useful constraints on the mass and location of the IMBH. The accelerations of all stars are consistent with zero within 3$\sigma$, but two stars have more than 2$\sigma$ acceleration measurements. We model both the velocity and the acceleration measurements to further constrain the IMBH properties (Methods); this calculation increases the lower limit on the black hole mass to 21,100 $M_\odot$ (99% confidence) and gives a preferred position for the IMBH 0.77″ northeast of the AvdM10 centre.

Apart from these constraints that are purely based on the assumed escape velocity and our astrometric data of the five robustly measured fast-moving stars, we also compared the full velocity distribution observed in the inner 10″ of ω Cen to already existing state-of-the-art N-body models[5] with various IMBH masses. Models with no IMBH, a stellar mass black hole cluster or an IMBH with a mass greater than 50,000$M_\odot$ are all strongly ruled out, whereas models with an IMBH mass of 39,000$M_\odot$ and 47,000$M_\odot$ are most consistent with the fraction of fast-moving stars and the observed velocity distribution. However, we caution that our comparisons show that low number statistics limit these comparisons, and mismatches with the overall velocity distribution suggest a need for improved modelling (see Methods for more details).

The detection of fast-moving stars in the centre of ω Cen strengthens the evidence for an IMBH in this cluster. Owing to the probable origin of ω Cen as the nucleus of the Gaia–Enceladus–Sausage dwarf galaxy[20,21], this black hole provides an important data point in the study of black hole demographics in low-mass galaxies, along with other black holes that have been detected in more massive globular clusters and stripped nuclei around M31 such as G1 ($M \approx 20,000M_\odot$) (refs. 22,23) or B023-G078 ($M \approx 100,000M_\odot$) (ref. 24). Moreover, this black hole provides the closest massive black hole and only the second after Sgr A* for which we can study the motion of multiple individual bound stellar companions. A comparison with the motion of

the stars in the Galactic centre is shown in the Methods and Extended Data Fig. 8.

A more precise estimate of the black hole mass requires dynamical modelling of all newly available kinematic data using models that include the impact of both an IMBH and mass-segregated dark remnants. The exact properties of the orbits of the fast-moving stars have to be determined by deep, pin-pointed follow-up observational studies. Spectroscopic observations with integral-field-unit instruments such as VLT MUSE[25] or JWST NIRSpec IFU[26] could yield LOS velocities for the fast-moving stars. Even more precise and deeper astrometric measurements with existing (VLTI GRAVITY+[27], JWST NIRCam[28]) or future (ELT MICADO[29], VLT MAVIS[30]) instruments could enable the detection of additional tightly bound stars, and the measurements of accelerations, key for obtaining direct measurements of the black hole mass. Our results also motivate revisiting the other likely accreted nuclear star clusters of the Milky Way[21], with M54 being the clearest case. For the search for IMBHs in other globular clusters, our results indicate that it may be necessary to extend kinematic studies to the faintest stars, which is observationally challenging for clusters at larger distances and with high central densities.

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

# Methods

## Discussion of previous IMBH detections in ω Centauri

The debate about an IMBH in ω Cen dates back almost two decades but has remained controversial. Early dynamical modelling based on LOS integrated-light velocity dispersion measurements suggested an IMBH mass of $M_{IMBH} = (4.0^{+0.75}_{-1.0}) \times 10^4 M_\odot$ (ref. 3). These results were challenged with a precise redetermination of the centre of the cluster[6] and dynamical modelling of proper motions[7] measured from multi-epoch HST imaging observations that placed an upper limit of $1.2 \times 10^4 M_\odot$ on the IMBH. Using additional integrated light observations and a centre based on the maximum LOS velocity dispersion, a best-fit IMBH mass of $M_{IMBH} = (4.7 \pm 1.0) \times 10^4 M_\odot$ was obtained in ref. 4. When assuming the AvdM10 centre, the IMBH mass was slightly lower, $M_{IMBH} = (3.0 \pm 0.4) \times 10^4 M_\odot$.

Subsequent comparisons of both proper motions and LOS velocities to N-body simulations continued to show evidence for an approximately $4.0 \times 10^4 M_\odot$ IMBH[31,32]. However, these observations were also shown to be fully consistent with a dark cluster of stellar mass black holes in the central region of ω Cen[8]. The lack of fast-moving stars in previous proper-motion catalogues supported this scenario over an IMBH[9]. Other works noted the influence of radial velocity anisotropy on dynamical mass estimates[33,34].

Most recently, the discovery of a counter-rotating core using VLT MUSE LOS velocity measurements of individual stars[35] highlighted once again the kinematic complexity of the centremost region of ω Cen. The centre of this counter-rotation coincides with the AvdM10 centre within about 5″ but is incompatible with the centres used in refs. 3,4.

## Previous accretion constraints in context

With the detection of a $10^{4-5} M_\odot$ IMBH, the upper limits on any accretion signal at X-ray[36] and radio[37] wavelengths make this the most weakly accreting black hole known. Deep, 291 ksec Chandra observations place an upper limit of the 0.5–7 keV luminosity of around $10^{30}$ ergs s$^{-1}$ (ref. 36), roughly 12 orders of magnitude below the Eddington limit. The radio upper limit implies an even fainter source, with the 5 GHz upper limit of $1.3 \times 10^{27}$ ergs s$^{-1}$ (ref. 37) corresponding to an implied X-ray luminosity using the fundamental plane of about $10^{29}$ ergs s$^{-1}$ (ref. 38). Assuming standard bolometric corrections of about 10 (ref. 39), this X-ray luminosity upper limit suggests an Eddington ratio of $\log(L_{bol}/L_{edd}) < -12$, far fainter than that for Sgr A*[40] or any other known black hole. This faint signal could be because of a combination of low surrounding gas density, a low accretion rate of that gas and/or a low radiative efficiency[37]. Low-luminosity active galactic nuclei including Sgr A* are brightest at infra-red and sub-millimetre wavelengths most likely because of synchrotron emission from compact jets[41,42]. Therefore, future observations with the James Webb Space Telescope or the Atacama Large Millimeter array would provide the highest sensitivity to any emission from the IMBH of ω Cen. Any detection would reveal the location of the IMBH as well as provide valuable constraints on the black hole accretion in this extremely faint source.

## Proper-motion measurements and sample selection

Our proper-motion measurements are based on the reduction of archival HST data of the central region of ω Cen, taken over a time span of more than 20 years. We used the state-of-the-art photometry tool KS2 (ref. 32) for the source detection and the astro-photometric measurements, and the established procedure described in refs. 43–46 to measure proper motions relative to the bulk motion of the cluster. The result of this extensive study is a proper-motion catalogue with high-precision measurements for 1.4 million stars out to the half-light radius of ω Cen with a typical temporal baseline of more than 20 years. Owing to the large number of observations (in total we reduced over 500 images and some stars in the central region have up to 467 individual astrometric measurements) the catalogue reaches unprecedented depth

and precision. The highest precision is achieved in the well-covered centre of the cluster, in which our proper motions have a median error of only about 6.6 μas yr$^{-1}$ (0.17 km s$^{-1}$) per component for bright stars.

The catalogue is larger than any other kinematic catalogue published for a globular cluster and significantly extends previous proper-motion catalogues for ω Cen[6,32,47]. A detailed comparison with other proper-motion datasets is published along with the catalogue[14]. In a following section and in Extended Data Fig. 3, we compare the completeness of the different catalogues to show that it is plausible that the fast-moving stars have been missed in previous searches.

We use a high-quality subset of the proper-motion catalogue to search for real fast-moving stars and limit spurious astrometric measurements (for example, two sources that are falsely identified as one) that can have apparent high proper-motion measurements. Our criteria for this subset are based on the amount of available data for the measurements. Specifically, we used only sources that had at least 20 astrometric measurements covering a temporal baseline of at least 20 years and a fraction of rejected measurements (based on sigma clipping) of less than 15%. We also made cuts on the quality of the proper-motion fit requiring both a proper-motion error less than 0.194 mas yr$^{-1}$ ≈ 5 km s$^{-1}$ and a reduced $\chi^2 < 10$ for the linear proper-motion fit for both the right ascension (RA) and declination (Dec) measurements. Apart from these quality selections, we also required the star to lie on the CMD sequence in an HST-based CMD (Fig. 2). These cuts help to drastically reduce the number of contaminants. These criteria are met uniformly out to a radius of about 90 arcsec; at larger radii they lead to selection effects due to reduced observational coverage. A total of 157,320 out of 241,133 (65.2%) entries of the proper-motion catalogue within $r < 90″$ match the combined criteria. Extended Data Table 1b shows the individual measured proper-motion components for the seven fast-moving stars. We note that for this analysis we have not applied the local a posteriori proper-motion corrections provided with the catalogue[14], as we are studying the central region that is well dithered and observed with various rotation angles. We verified that applying these corrections would neither change our fast-moving star sample nor alter our conclusions.

## Details on verification for fast-moving stars

The criteria detailed above should lead to a clean dataset with very few spurious proper-motion measurements. To ensure that the measurements for the fast-moving stars are reliable, we inspected each of them carefully.

As a first step, we tested the quality of the raw astrometric measurements by studying several goodness-of-fit parameters and photometric quality indicators for the point-spread-function fits used to measure stellar positions (Extended Data Fig. 3). We performed this analysis for the WFC3/UVIS F606W filter as it is the most used filter in the centre of ω Cen, and each star has at least 195 measurements in this filter. To verify the goodness of fit, we used the mean of the so-called quality-of-fit (QFIT) flag and the radial excess value, both of which take into account the residuals of the point-spread-function fit. Furthermore, we looked at the mean of the ratio of source flux with respect to the flux of neighbouring sources within its fit aperture. All five stars used for our analysis behave typically for well-measured stars of their magnitude and none of them show extreme values that would indicate problems with the photometry. It is noteworthy that the two stars excluded from our analysis based on their velocities being less than $3\sigma$ above the escape velocity show some deviations. Star B has a relatively low mean QFIT value and high radial excess. Star G is the only one in the sample for which the neighbour flux to star flux ratio is larger than 1.

As a second step, we looked at the stars in several stacked HST images taken at various epochs ranging from 2002 to 2023. The extensive multi-epoch imaging is demonstrated in Extended Data Fig. 2. The proper motion of a star at the escape velocity (62 km s$^{-1}$) is 2.41 mas yr$^{-1}$. Therefore, we expect to see a displacement of at least 50 mas over 21 years, which corresponds to 1.25 WFC3/UVIS pixels. This motion can

be seen by eye for all seven stars in the multi-epoch images. Again the excluded stars B and G stick out, that is, that they are partially blended with the neighbouring stars, thus explaining their larger astrometric errors.

Finally, we tested the reliability of our proper-motion measurements by limiting the raw position measurements to different subsets and redoing both the linear and quadratic fits to the motion of the stars. The first test run included only high signal-to-noise measurements. The second test run included only measurements taken with the WFC3/UVIS F606W filter. By using only one filter, we are immune to colour-induced effects such as a partially resolved blend between two differently coloured stars. The proper motions of all fast-moving stars are consistent within the measurement uncertainties using both methods.

## Comparison with other proper-motion datasets

Before our analysis, two other high-precision proper-motion catalogues based on HST data have been published[6,32] covering the centre of ω Cen. Both datasets were searched for central high proper-motion stars, but none of the stars in our sample have been reported before. To understand why this is the case, we compare the completeness of the different catalogues. Extended Data Fig. 3a shows histograms of the magnitudes of stars with measured proper motions. In the inner 20″, our new catalogue contains more than three times the number of stars of the literature catalogues and extends to significantly fainter magnitudes. The newly detected fast-moving stars all lie at faint magnitudes, in which the completeness of the older catalogues is significantly lower than in the new proper-motion catalogue. This is because of the larger amount of data and the updated source-finding algorithms in the new catalogue, which explains the previous non-detection of the fast-moving stars.

## Discussion of photometric errors

Apart from the astrometric reliability, we also studied the quality of the photometric measurements used to locate the stars in the CMD. Although the innermost stars are faint, their statistical photometric errors are small because of the large number of individual photometric measurements combined to a weighted mean value. The statistical errors range from 0.004 mag to 0.037 mag and are given in Extended Data Table 1c. However, especially for faint stars, this statistical error is not able to capture systematic issues caused, for example, by the influence of brighter neighbouring stars. These issues can be identified only by verifying the quality of the point-spread-function fit used to determine the individual photometric measurements. We report the mean quality of fit, radial excess and neighbour flux to source flux ratio flags for both filters in Extended Data Table 1c and compare them with those of stars at similar ($\Delta m < 0.5$) magnitudes in Extended Data Fig. 3. All stars in the robustly measured sample show typical quality of fit for their respective magnitude. We note that stars B and G (which were excluded from the analysis) show comparatively poor QFIT. Stars E and G show a possible flux contribution from a neighbouring source (indicated by a high radial excess value and a high neighbour flux to source flux ratio). This can be confirmed by the stacked images shown in Extended Data Fig. 2, in which these stars show a close neighbour. Owing to the relatively bright magnitude of star E and the low astrometric scatter, we still consider its measurement valid.

## Density of Milky Way contaminants versus fast-star background

To quantify our expected level of contamination from Milky Way foreground and background stars, we compared our results with those of a Besançon model[18]. Using the m1612 model, we simulate a 1 square degree patch centred on ω Cen retrieving Johnson colours and kinematics. We then transform the model Johnson $V$ and $I$ magnitudes into F606W and F814W magnitudes using linear relations fitted to Padova models[48] between $V–I$ of 0 and 2. We use the same colour cuts and consider stars between F606W of 16–24. We then count the number of stars with a total proper motion above 2.41 mas yr$^{-1}$ (equivalent to

our velocity cutoff at the escape velocity $v_{esc} = 62$ km s$^{-1}$). These would appear as contaminants in our fast-moving star sample. We find a density of 0.0039 stars per arcsec$^2$. This is somewhat higher than the $0.0026 \pm 0.0003$ stars per arcsec$^2$ found as the background level in our observations. This discrepancy is alleviated by considering only 65.2% of all stars within our catalogue meet the high-quality criteria used for our fast-moving star selection (see above). Correcting for this factor, we get an expected background of 0.0025 stars per arcsec$^2$, perfectly matching the observed background density (Extended Data Fig. 1). This suggests that our background level is consistent with being predominantly Milky Way contaminants. Relaxing our requirement that stars be more than $3\sigma$ above the escape velocity results in a higher observed background level of 0.0042 stars per arcsec$^2$, no longer consistent with the Milky Way background. This suggests that our stricter definition of a fast-moving star reduces contamination from poorly measured stars in ω Cen to a negligible level.

## Other scenarios that could explain the fast-moving stars

A complete contamination of our sample by Milky Way foreground and background stars that are non-members of ω Cen can be ruled out statistically. We now explore and rule out alternative scenarios to the fast-moving stars being bound to an IMBH. One alternative explanation for stars with a high velocity is to have them bound in a close orbit with a stellar-mass black hole. This scenario can be ruled out for BHs less than $100 M_\odot$, as the periods required to reach the observed velocities are less than 10 years, which is well within the 20-year span over which we have observed linear motions.

Another scenario could be that the stars are actually unbound from the cluster and have recently been accelerated by three-body interactions, either with stellar-mass black hole binaries or with an IMBH. Ejection by an IMBH in the centre of the cluster can be ruled out by the very high rate of ejections necessary to sustain the observed number of fast-moving stars within the centre and the absence of observed fast-moving stars at larger radii. To sustain a density of 0.18 fast-moving stars per arcsec$^2$ in the inner 3 arcsec (equivalent to our conservative sample of five stars) moving with at least 2.4 mas yr$^{-1}$ would require ejections with a rate of 0.004 stars per year. This would lead to about 117 additional fast-moving stars at larger radii (20″ < $r$ < 90″), apart from around 60 foreground stars expected from the (completeness corrected) Besançon Milky Way model. In our dataset, we find 61 fast-moving stars between 20″ < $r$ < 90″, consistent with the expected Milky Way background but not consistent with a substantial number of additional ejected stars. Moreover, a high hypothetical ejection rate of 0.004 stars per year would deplete all of the about 10 million stars of ω Cen in just 2.5 Gyr. If no IMBH is present, accelerations of stars above the escape velocity are still possible by three- or four-body interactions between stellar or compact object binaries[49]. However, these interactions would not be limited to the innermost few arcseconds of the cluster because of the slowly varying stellar density in the core of ω Cen. Furthermore, the expected rate of these ejection events is of the order of less than one ejection per 1 million years, around 1,000 times lower than needed to explain the observed number of fast-moving stars in the centre of ω Cen[49,50].

## Search for the fast-moving stars in recent LOS velocity data

Line-of-sight (LOS) velocities of the fast-moving stars could help to exclude contaminants and provide further constraints on the orbits of the stars and the mass and position of the IMBH. The deepest and most extensive spectroscopic catalogue of stars in ω Cen is part I of our recently published oMEGACat[13]. This catalogue was created using a large mosaic of observations with the VLT MUSE integral field spectrograph and contains both LOS velocity measurements and metallicities for more than 300,000 stars within the half-light radius of ω Cen. Although we could successfully cross-match five of the seven fast-moving stars, their signal-to-noise ratio is typically too low (about 2)

for reliable velocity measurements, in particular for the four fastest, innermost stars.

We could, however, obtain an LOS velocity value for star E ($v_{LOS}$ = 261.7 ± 2.7 km s$^{-1}$) and star F ($v_{LOS}$ = 232.5 ± 4.0 km s$^{-1}$). These velocities are very close to the systemic LOS velocity of ω Cen (232.99 ± 0.06 km s$^{-1}$; ref. 13), confirming their membership in the cluster, as the Milky Way foreground is centred at $v_{LOS} \approx 0$ with a dispersion of 70 km s$^{-1}$ (ref. 18). However, as the relative LOS velocity with respect to the cluster is low and we have only those two velocities for these outer stars, the LOS velocities do not add stronger constraints on the IMBH. For this reason, we did not include them in the rest of our analysis.

### Testing the robustness of the assumed escape velocity

**Varying the parameters of the *N*-body models.** Because we use the escape velocity of ω Cen (assuming no IMBH is present) as the threshold for determining whether a star is considered fast or not, it is important to verify the robustness of the escape velocity value. We adopt an escape velocity $v_{esc}$ = 62 km s$^{-1}$ (ref. 16); we have verified this value based on fitting similar *N*-body models to several state-of-the-art datasets, including MUSE LOS velocity dispersion measurements[51] and HST proper-motion-based dispersion measurements[52] for the central kinematics and Gaia DR3[53] measurements at larger radii using an assumed distance of 5.43 kpc. We varied both the assumed initial stellar mass function (using either the canonical Kroupa IMF[54] or the bottom-light IMF derived in ref. 55) and the black hole retention fraction (assuming values of 10%, 30%, 50% or 100%). Despite changes to the central mass-to-light (*M/L*) ratio between the models, the central escape velocity changes only minimally, with a range of values from 61.1 km s$^{-1}$ to 64.8 km s$^{-1}$. Adopting any of these values leaves our sample of seven central stars above the escape velocity unchanged.

**An independent test using surface-brightness profiles.** As a second test, independent of the *N*-body models, we calculated an escape velocity profile based on a surface brightness profile using various surface brightness profiles and dynamical models from the literature. We started by parameterizing the surface brightness profile using multi-Gaussian expansion (MGE)[56] models. We then converted the surface brightness to a mass density using several mass-to-light ratios and distances used in the literature. From the mass density, we can derive the gravitational potential ($\Phi(r)$). The escape velocity profile is then given by

$$v_{esc}(r) = \sqrt{2(\Phi(r_{tidal}, 0) - \Phi(r, 0))}$$

with the tidal radius $r_{tidal}$ = 48.6′ ≈ 74.6 pc from refs. 57,58.

These tests showed that the central escape velocity does not depend strongly on the stellar mass distribution in the centremost region, instead it is dominated by the global *M/L* ratio and the assumed distance. The early dynamical models in the IMBH debate assumed both a distance of 4.8 kpc (ref. 59) and an *M/L* of 2.6 (vdMA10; ref. 7) or 2.7 (N08; refs. 3,4). With these values our tests give a central escape velocity of 55.4 km s$^{-1}$ (vdMA10) and 56.9 km s$^{-1}$ (N08). If we would also use the 4.8 kpc distance to scale the proper motions, this gives a cutoff of 2.43 mas yr$^{-1}$ (vdMA10) and 2.48 mas yr$^{-1}$ (N08), close to the adopted cutoff at 2.41 mas yr$^{-1}$ and not changing the sample of the seven fast-moving stars detected. Owing to the parallax measurements of the Gaia satellite and updated kinematic distance measurements, the distance to ω Cen was robustly redetermined and larger values have been found (5.24 ± 0.11 kpc; ref. 60; 5.43 ± 0.05; ref. 12). A dynamical model using the same surface brightness profile as ref. 3 but a larger distance of 5.14$_{-0.24}^{+0.25}$ kpc was presented in ref. 8; this study found an *M/L* of 2.55$_{-0.28}^{+0.35}$. Using these values, the central escape velocity derived from the surface brightness profile is 61.1 km s$^{-1}$ (equivalent to a proper motion of 2.51 mas yr$^{-1}$); again not changing our fast-moving stars sample. Finally, varying the distance of any model by 0.2 kpc while holding the *M/L* constant results in variation in the escape velocity of the order of ±3 km s$^{-1}$. These results show that our fast-moving star limit ($v_{esc}$ = 62 km s$^{-1}$ at a distance of 5.43 kpc) is consistent with the escape velocity values directly derived from surface-brightness profiles and several dynamically estimated *M/L* values. To visualize the escape velocity, we calculated the escape velocity using the surface-brightness profile in ref. 3, an *M/L* of 2.4, and a distance of 5.43 kpc as found from the *N*-body models with a cluster of stellar mass black holes. The resulting profile is shown in Extended Data Fig. 5f. The predicted escape velocity is flat out to approximately 50″, a property shared by all of the calculated escape velocity profiles. This makes the detection of the fast-moving stars only in the central few arcseconds more compelling.

**An empirical confirmation of the central escape velocity.** We make one final, and relatively model-independent, empirical confirmation of the central escape velocity based on the distribution of 2D velocities in the innermost region of ω Cen (Extended Data Fig. 4). As we have seen in the analysis above, the escape velocity varies only slightly within the inner about 50″ of the core of ω Cen. Furthermore, the velocity dispersion profile is relatively flat in the innermost 10″, with a value of about 20 km s$^{-1}$ (refs. 6,35,52). Therefore, we would expect rather similar distributions of stellar velocities in both the centre (0″ < $r$ < 3″) and an outer ring at (3″ < $r$ < 10″). Although we observe a clear excess of fast-moving stars in the inner 3 arcsec, there is a sharp cutoff very close to the adopted escape velocity in the (3″ < $r$ < 10″) bin. Although there is a total of 2,090 stars, there is only one star with a velocity significantly faster than the escape velocity (instead of 17 stars expected from a 2D Maxwell–Boltzmann distribution with $\sigma_{1D}$ = 20 km s$^{-1}$). This suggests that the stars with these velocities have escaped the central region. This outer fast-moving star has a 2D velocity of 75.8 km s$^{-1}$ and is at a radius of $r$ = 9.5″. From the density of Milky Way contaminants with apparent velocities above the escape velocity, we would expect about 0.7 foreground stars in the (3″ < $r$ < 10″) region; therefore, this fast-moving star is consistent with being a Milky Way foreground star.

### The escape velocity provides a minimum black hole mass

The escape velocity for an isolated black hole is given by

$$v_{esc,BH} = \sqrt{\frac{2GM_{BH}}{r_{3D}}}.$$

In ω Cen, we have to take into account the potential of the globular cluster as well. If we assume this to be constant over the very small region in which we found the fast-moving stars (an assumption that agrees with the published surface brightness profiles, see Extended Data Fig. 6f), we obtain

$$v_{esc,total} = \sqrt{v_{esc,BH}^2 + v_{esc,cluster}^2}$$

If a star at the distance of $r_{3D}$ with a velocity $v_{3D}$ is bound to the black hole, we can calculate the following lower limit on the black hole mass:

$$M_{BH} > \frac{(v_{3D}^2 - v_{esc., cluster}^2)r_{3D}}{2G} \geq \frac{(v_{2D}^2 - v_{esc., cluster}^2)r_{2D}}{2G}$$

A lower limit can also be calculated if the LOS velocity and distance are not known, as $v_{3D} \geq v_{2D}$ and $r_{3D} \geq r_{2D}$. As we do not know the exact 2D position of the black hole relative to the fast-moving stars, we calculated this lower limit for all stars and a grid of assumed 2D locations around the AvdM10 centre[6]. Each individual star alone would allow for a very low mass, as the location of the black hole could coincide with the star (Extended Data Fig. 5a–d). However, combining these limits for all stars gives a higher minimum black hole mass (Extended Data Fig. 5e). If we assume that all five robustly detected stars are bound to the black hole, the lower limit is about 8,200 $M_\odot$ and the minimum mass

location is only 0.3″ away from the AvdM10 (ref. 6) centre at the location RA = 201.6966908° and Dec = −47.4795066°. If we assume that the two most constraining stars are just random foreground contaminants, which is ruled out at the 3σ level (P = 0.0026), this limit drops to about 4,100$M_\odot$, still well within the IMBH range.

## Acceleration measurements

The astrometric analysis in the catalogue[14] considered only linear motions of the stars. If there is a massive black hole present near the centre, we might also be able to measure accelerated motion of the closest stars, allowing for a direct mass measurement of the black hole. With an IMBH mass of 40,000$M_\odot$ and at a radius of 0.026 pc (1″ on the sky), the acceleration of a star would be 0.25 km s⁻¹ yr⁻¹ (or 0.01 mas yr⁻²). This is at the limit of the precision of our current dataset. With a 20-year baseline, we expect only a deviation of 0.05 pixel from a linear motion. For bright stars, the astrometric uncertainty can be as low as 0.01 pixel; however, for the faint fast-moving stars we have detected, the errors are significantly larger.

To constrain these possible accelerations, we repeated the fit of the motion of each star enabling the addition of a quadratic component. The results for this fit are shown in Extended Data Table 1. The accelerations of the stars are consistent with zero within 3σ, but two stars have more than 2σ acceleration measurements. The errors on our acceleration measurements lie between 0.004 mas yr⁻² and 0.03 mas yr⁻² and are, therefore, of a magnitude similar to the expected acceleration signal. The strongest acceleration is shown by star B, which has been excluded from the robust subset of fast-moving stars because its proper motion is not 3σ above the escape velocity. Owing to the proximity of a bright neighbour star, we do not deem this acceleration measurement to be reliable.

As the LOS distances to the fast-moving stars are unknown, it is not possible to place direct constraints on the IMBH mass using the upper limits on accelerations. If no acceleration is detected, as is the case for the centremost star A, this could mean that either the black hole is not very massive or the LOS distance of star A to the black hole is large. Combining the measurements for the ensemble of fast-moving stars and making some assumptions on their spatial distribution still enables us to use the acceleration limits to place further constraints on the black hole mass and its location. This is described in the next section.

## Markov chain Monte Carlo fitting of the acceleration data

Assuming the fast-moving stars are bound to the IMBH, we can model the stars as being on Keplerian orbits around the IMBH. We used Bayesian analysis to sample the posterior distribution for the unknown mass and position of the black hole. In this analysis, there were eight free parameters: black hole mass; its on-sky x- and y-positions; and five LOS distances between the black hole and each fast-moving star. This analysis makes use of the available astrometric observations but stops short of modelling individual stellar orbits that would introduce more free parameters.

We use a likelihood function with these eight free parameters and give the likelihood based on the observed on-sky x- and y-acceleration, proper motion and position of the five robustly measured fast-moving stars. For each star, we calculated a first likelihood term based on the modelled acceleration $a_{modelled}$ using a Gaussian distribution with mean $a_{observed}$ and width equal to the acceleration uncertainty. The second term in the likelihood accounts for the escape velocity constraints and is kept constant if the observed 2D velocity of the star is below the modelled escape velocity. For stars with 2D velocities above the modelled escape velocity, the likelihood is a Gaussian distribution with mean $v_{2D} - v_{esc\,total}$ and width equal to the uncertainty in observed proper motion.

We make these assumptions about the model: (1) The black hole mass is between 1$M_\odot$ and 100,000$M_\odot$, because a black hole mass beyond this upper limit is ruled out by our N-body models. (2) The black hole is located within the distribution of the fast-moving stars. We use a Gaussian prior in the black hole x- and y-positions with a mean equal to the mean position of the fast-moving stars and width equal to their one-dimensional positional standard deviation, $\sigma_{stars} = 0.0221$ pc; we also use a cutoff at ±0.16 pc. (3) The stellar positions are isotropically distributed around the black hole. We model the LOS positions of the stars relative to the black hole using a Gaussian distribution with mean 0 and width $\sigma_{stars}$.

The posterior was sampled using a Markov chain Monte Carlo (MCMC) ensemble sampler implemented using the package emcee[61] using recommended burn-in and autocorrelation corrections. We show the posterior distribution for the black hole mass in Extended Data Fig. 6a. The 99% confidence lower limit (21,100$M_\odot$) is significantly higher than that derived from escape velocity constraints alone, whereas the upper limit on the mass is not well constrained. We also find a position for the black hole east of the AvdM10 centre, with $\Delta x = -0.017^{+0.017}_{-0.031}$ pc and $\Delta y = 0.011^{+0.011}_{-0.025}$ pc (Extended Data Fig. 6b). The coordinates of the MCMC based centre estimate are RA = 201.6970988° and Dec = −47.4794533°. We note that the black hole location estimate is dominated by the marginal 2σ acceleration signal of star D, which is the faintest star in the sample; follow-up studies are required to obtain more precise acceleration measurements.

## N-body models

Apart from the analysis of stars with velocities above the escape velocity, we also used a set of existing N-body models with and without central IMBHs to get further constraints on the IMBH mass. We compared the simulations to the full velocity dispersion and surface density profile of ω Cen to determine the best-fitting model and the mass of a central IMBH. The set of models and the details of the fitting procedure are described in detail in refs. 5,9. We note that these models have been presented already in the literature, but the fits to these models have been updated to incorporate the most recent Gaia DR3 data.

In short, the models started from King profiles[62] with central concentrations between c = 0.2 and c = 2.5 and initial half-mass radii between $r_h = 2$ pc and $r_h = 35$ pc. In the models with an IMBH, we varied the mass of the IMBH so that it contains either 0.5%, 1%, 2% or 5% of the cluster mass at T = 12 Gyr when the simulations were stopped. The models with an IMBH assumed a retention fraction of stellar-mass black holes of 10%, whereas in the models without an IMBH we varied the assumed retention fraction of stellar-mass black holes between 10% and 100%. At the end of the simulations, we calculated surface density and velocity dispersion profiles for each N-body model and then determined the best-fitting model by interpolation in our grid of models and using $\chi^2$ minimization against the observed velocity and surface density profile of ω Cen.

The velocity distributions from observations and the models are shown in Extended Data Fig. 7. We compare the distribution of measured 2D stellar velocities in the inner 10″ of ω Cen with the various models using a Kolmogorov–Smirnov test. Moreover, we compare the fraction of fast-moving stars in the innermost 3 arcsec (Extended Data Table 2). Models without an IMBH and with a 20,000$M_\odot$ IMBH are both strongly excluded by both the overall velocity distribution and the complete lack of fast-moving stars. The overall velocity distribution is in best agreement with the 47,000$M_\odot$ distribution, whereas the fraction of fast-moving stars is best matched by the 39,000$M_\odot$ simulation. This tension might be alleviated in future models that contain both an IMBH and a cluster of stellar mass black holes. We caution that these simulations have smaller numbers of stars than observed and that there can be substantial variations in the distribution of central stars due to strong encounters with remnants and binaries. Nonetheless, the simulations suggest that black holes with masses of $M \lesssim 50,000M_\odot$ are consistent with the observed distribution of central velocities and fast-moving stars, whereas the no-IMBH case and significantly more

massive black holes are disfavoured because of an overprediction of fast-moving stars. Updated *N*-body models fit to the oMEGACat kinematic data and dynamical modelling of these same datasets with Jeans models are currently underway.

## Comparison with S-stars in the Galactic centre

The black hole indicated by our fast-moving star detection is only the second after Sgr A*, for which we can study the motion of multiple individual bound stellar companions. Therefore, the extensively measured stars around Sgr A* provide a unique comparison point to our fast-moving star sample. We compare the motions of the stars in the S-star catalogue in ref. 17 with our ω Cen fast-moving star sample in Extended Data Fig. 8. When taking into account the different distances and the approximate black hole mass ratio of 100, the motions show similar amplitudes. However, the density of tracers in ω Cen is significantly lower, despite the greater depth of the observations.

## Data availability

The data used in this paper are based on archival observations taken with the HST that are freely available in the Mikulski Archive for Space Telescopes. All used observations have been grouped under a DOI (https://doi.org/10.17909/26QJ-G090)[63]. Moreover, the full proper-motion catalogue is made public along with the respective publication[14].

## Code availability

We used the following Python packages to perform the analysis: matplotlib[64], scipy[65], numpy[66], astropy[67] and emcee[61]. The *N*-body simulations were run with the publicly available NBODY6 code[68]. We can share the code used in the data analysis upon request.

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

**Acknowledgements** This study is based on the observations with the NASA/ESA HST, obtained at the Space Telescope Science Institute, which is operated by AURA, under NASA contract NAS 5-26555. S.K. acknowledges funding from UKRI in the form of a Future Leaders Fellowship (grant no. MR/T022868/1). A.S., M.W. and A.B. acknowledge support from HST grant GO-16777. A.B. acknowledges support from STScI grants GO-15857 and AR-17033. A.F.K. acknowledges funding from the Austrian Science Fund (FWF) (https://doi.org/10.55776/ESP542). M.A.C. acknowledges support from Fondecyt Postdoctorado, project no. 3230727. A.M. acknowledges funding from PRIN 2022 2022MMEB9W (principal investigator: A. F. Marino).

**Author contributions** All authors helped with the interpretation of the data and provided comments on the paper. M.H. has led the analysis of the data and is the main author of the text. A.S. and N.N. designed the overall project with significant contributions from A.B., G.v.d.V. and S.K.; A.S., N.N., H.B., M.W. and A.D. contributed to the text. A.S. determined the expected density of Milky Way contaminants. A.B., M.L. and J.A. provided their expertise on astrometric measurements with the HST. H.B. provided and fitted the *N*-body models for the analysis. M.W. ran the Bayesian analysis used to constrain the IMBH mass and position. A.D. performed the surface-brightness-profile-based calculations. S.K. and M.S.N. helped to find available LOS data for the stars.

**Funding** Open access funding provided by Max Planck Society.

**Competing interests** The authors declare no competing interests.

**Additional information**
**Correspondence and requests for materials** should be addressed to Maximilian Häberle.

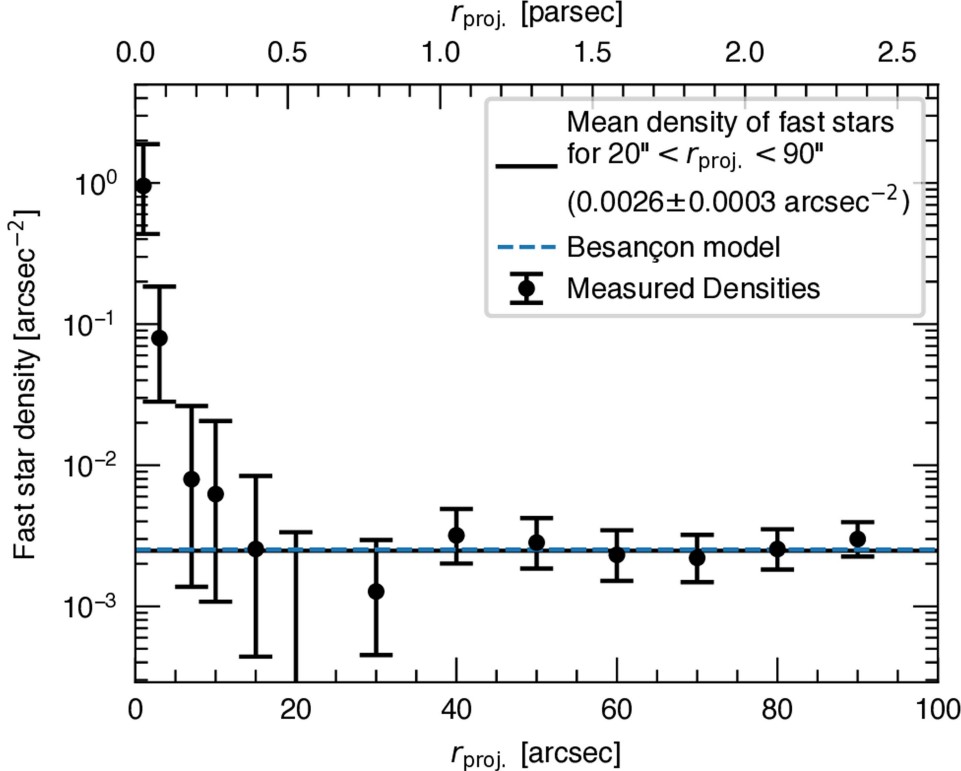

**Extended Data Fig. 1 | Number density of fast-moving stars.** The black markers with errorbars (1$\sigma$; based on Poisson statistics) show the measured number density of robustly detected fast-moving stars determined in radial bins with respect to $\omega$ Cen's center. A constant density of Galactic fore-/background stars is expected, but we see a strong and statistically significant rise of the density towards the AvdM10[6] centre. Based on the number density of fast-moving stars at large radii (which is consistent with predictions from a Besançon Milky Way model, dashed line), only ~0.073 fast-moving stars are expected within the central 3″ compared to the 5 observed stars.

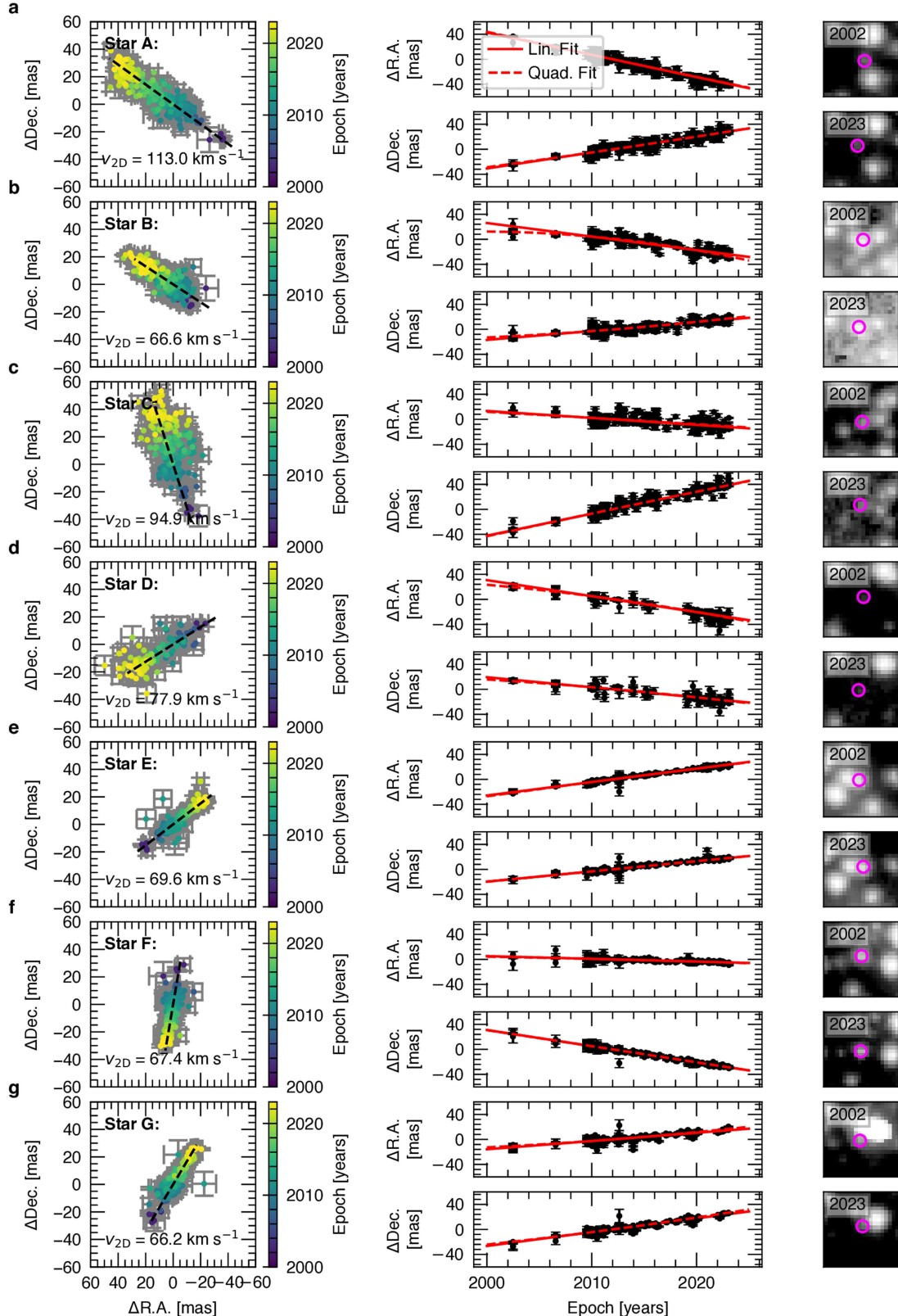

**Extended Data Fig. 2 | Astrometry and multi-epoch imaging for all fast-moving stars.** Each row (**a–g**) shows the astrometry and imaging for one of the seven fast-moving stars. The left column shows the raw astrometric measurements used to determine the proper motions, colour-coded by the epoch of their observation. The centre column shows the linear and quadratic fits to both the R.A. and Dec. position change of the stars. Errorbars correspond to the 1σ error on the individual position measurements. The right column shows stacked images from 2002 (ACS/WFC F625W) and 2023 (WFC3/UVIS F606W); the positions of the fast stars are marked with a pink open circle.

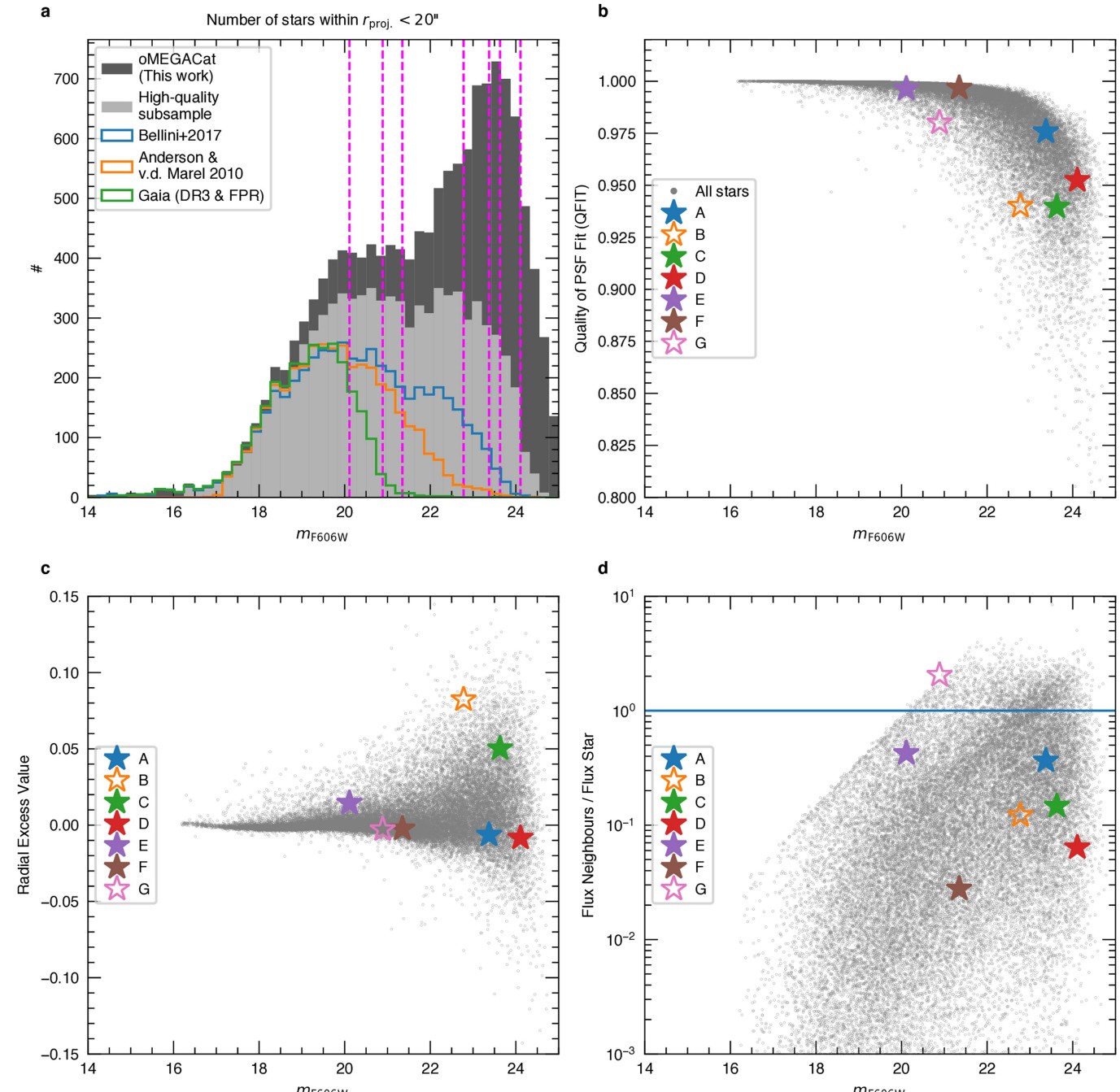

**Extended Data Fig. 3 | Completeness of the catalogue and photometric diagnostics.** Panel **a** compares the completeness of the various available proper motion datasets for the core of $\omega$ Cen, using histograms of the magnitude distribution. The new oMEGACat[14] has significantly higher completeness and reaches fainter magnitudes than the literature catalogues even if we apply the strict quality criteria used in this work. This explains why previous catalogues have not found the faint fast-moving stars (marked with vertical lines) we detect here. The grey dots in **b**–**d** show the mean of photometric diagnostics for the raw PSF photometry measurements for the fast-moving stars compared with the bulk of stars in the catalogue. The first panel shows the QFIT parameter, given by the linear correlation function between the PSF and the pixel values in the image. The second panel shows the radial excess parameter, a parameter that compares the residual flux inside versus outside of the fit aperture. The third panel shows the flux ratio between the flux of a star itself and its neighbouring sources. All five robustly measured stars show typical behaviour for their magnitude, while the excluded fast-moving stars (B, G) are influenced by bright neighbouring sources.

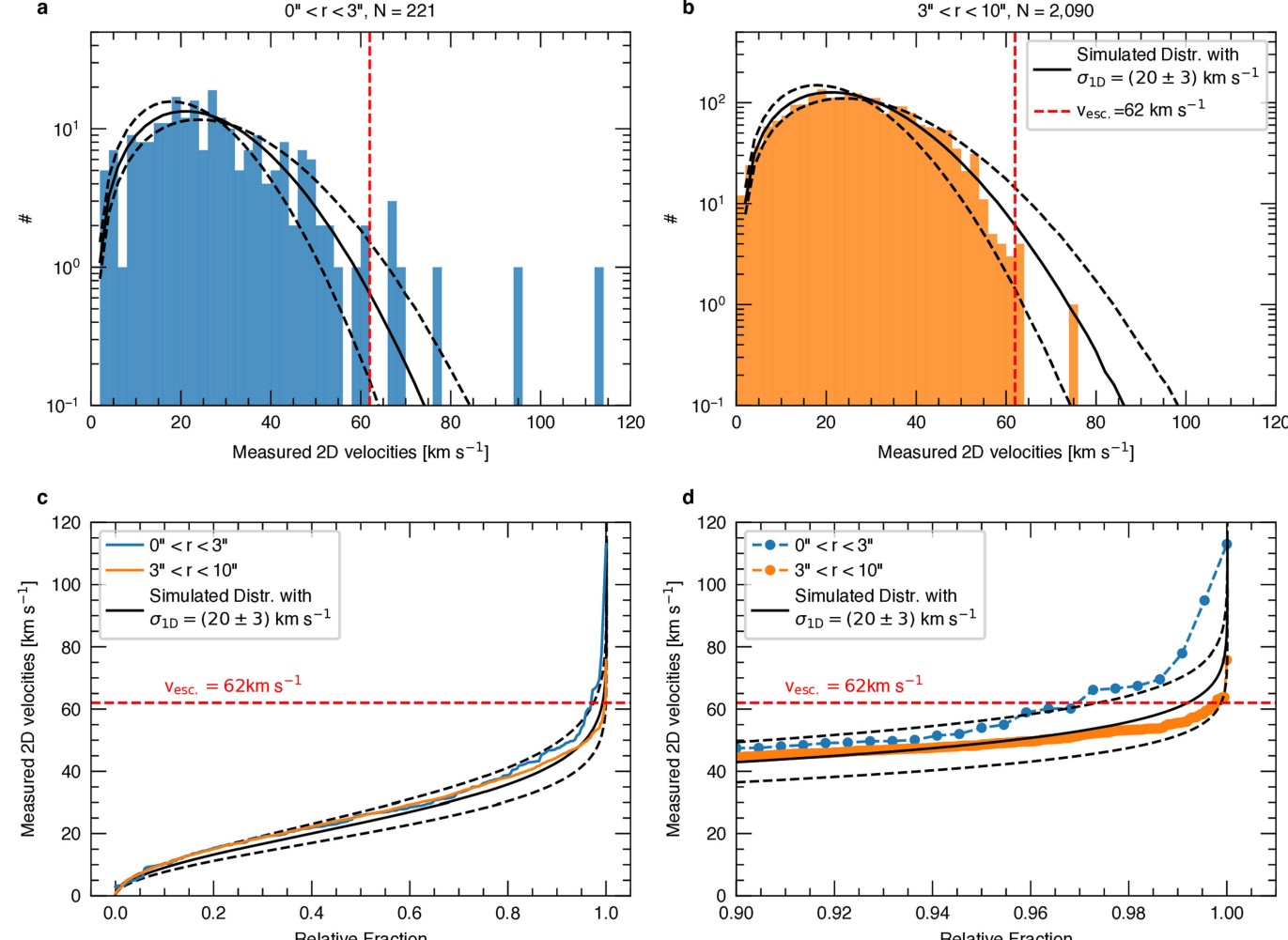

**Extended Data Fig. 4 | Empirical verification of the escape velocity.** Panels **a** and **b** show histograms of the observed 2D velocity distribution in the very centre (0″ < r < 3″; **c**) and in an outer ring (3″ < r < 10″; **d**). While the lower velocities are well described by a 2D Maxwell-Boltzmann distribution with $\sigma_{1D} = 20$ km s$^{-1}$ (marked with a solid black line, the dashed black lines refer to alternative distributions with $\sigma_{1D} = 17$ km s$^{-1}$ and $\sigma_{1D} = 23$ km s$^{-1}$), there are clearly notable differences at higher velocities. Those become especially visible in the cumulative normalized histogram shown in **d** and the zoom-in in **e**: while the distribution between (0″ < r < 3″, blue line) shows an excess of fast-moving stars, the distribution at larger radii (3″ < r < 10″, orange line) shows a clear deficit of stars at velocities larger than the escape velocity, making the used escape velocity threshold very plausible. Even though the sample is ten times larger, there is only a single star with a velocity significantly larger than $v_{esc}$. This star has a 2D velocity of 75.8 km s$^{-1}$ and is at a radius of r = 9.5″. It is consistent with being a Milky Way foreground star.

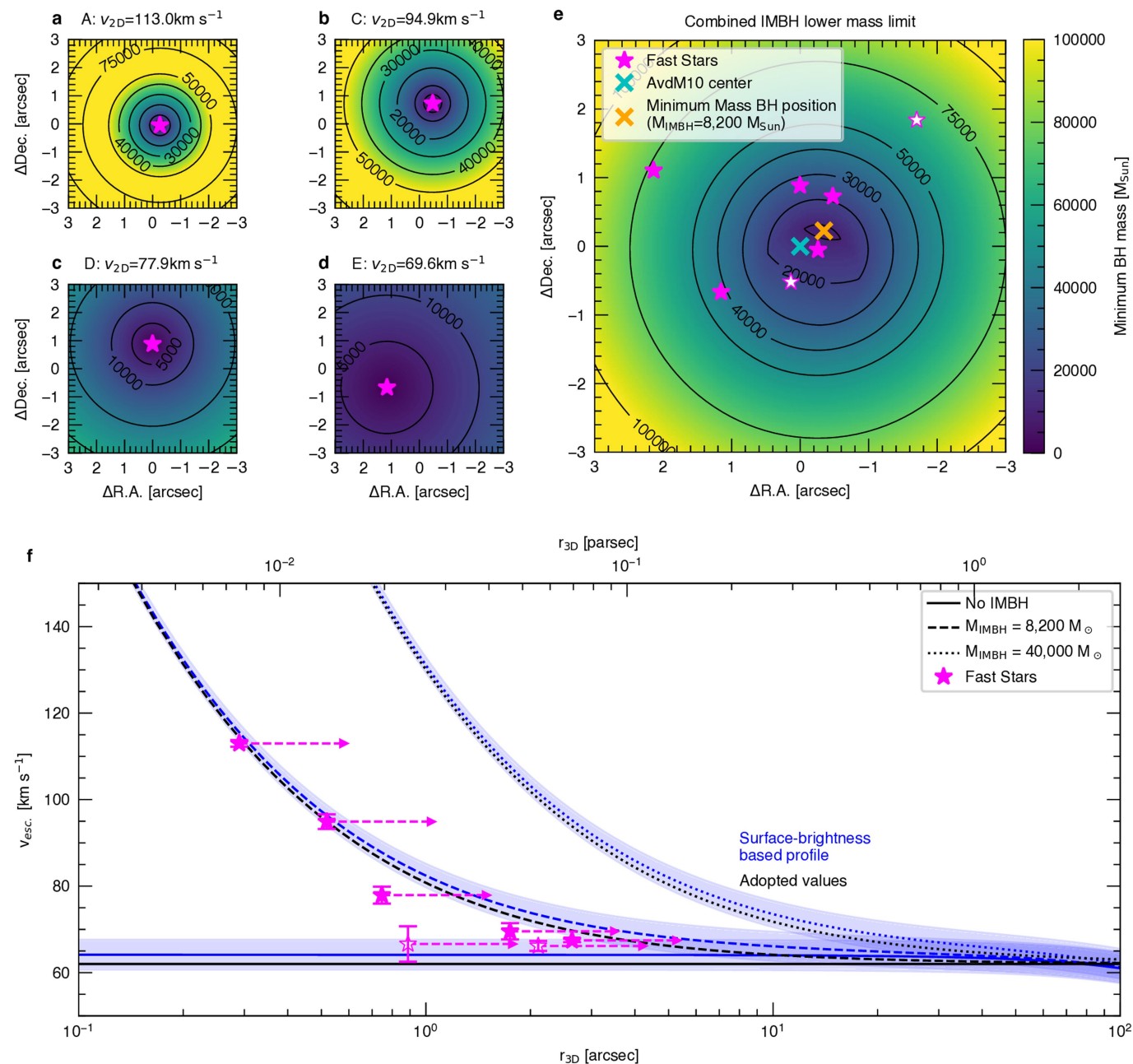

**Extended Data Fig. 5 | Determination of a lower limit on the IMBH mass using the escape velocity.** The presence of stars with velocities above the escape velocity of the cluster indicates that they are bound to a massive object. Since we neither know the mass nor the exact position of the object, we can only infer a lower mass limit for each possible 2D location. The four left plots (**a**–**d**) show the contours of the minimum black hole mass indicated by the four centremost robustly measured fast-moving stars. By combining the minimum black hole mass constraints from each of the fast-moving stars, we can find the position that allows for the lowest IMBH mass (**e**). This analysis indicates a firm lower limit of around 8,200 $M_\odot$. Our minimum mass location only differs by ~0.3 arcsec from the AvdM10[6] centre. This result does not significantly change if we assume that some of the fast-moving stars are contaminants and remove them from the analysis. Panel **f** shows the results for a surface brightness profile based escape velocity profile in blue, using the surface brightness profile from[3] and the dynamical distance (5.43 kpc) and M/L ratio (2.4) derived from N-body models, either without any IMBH, with a 8,200 $M_\odot$ IMBH, or a 40,000 $M_\odot$ IMBH. The radii are measured with respect to the minimum mass centre shown in **e**. The shaded regions indicate the uncertainty introduced by an assumed error of ±0.2 kpc on the distance. The surface-brightness profile based escape velocity is compatible with the adopted value of $v_{esc.}$ = 62 km s$^{-1}$. The profile without an IMBH is also nearly flat in the inner ~50″, justifying the assumption of a flat profile in the innermost region.

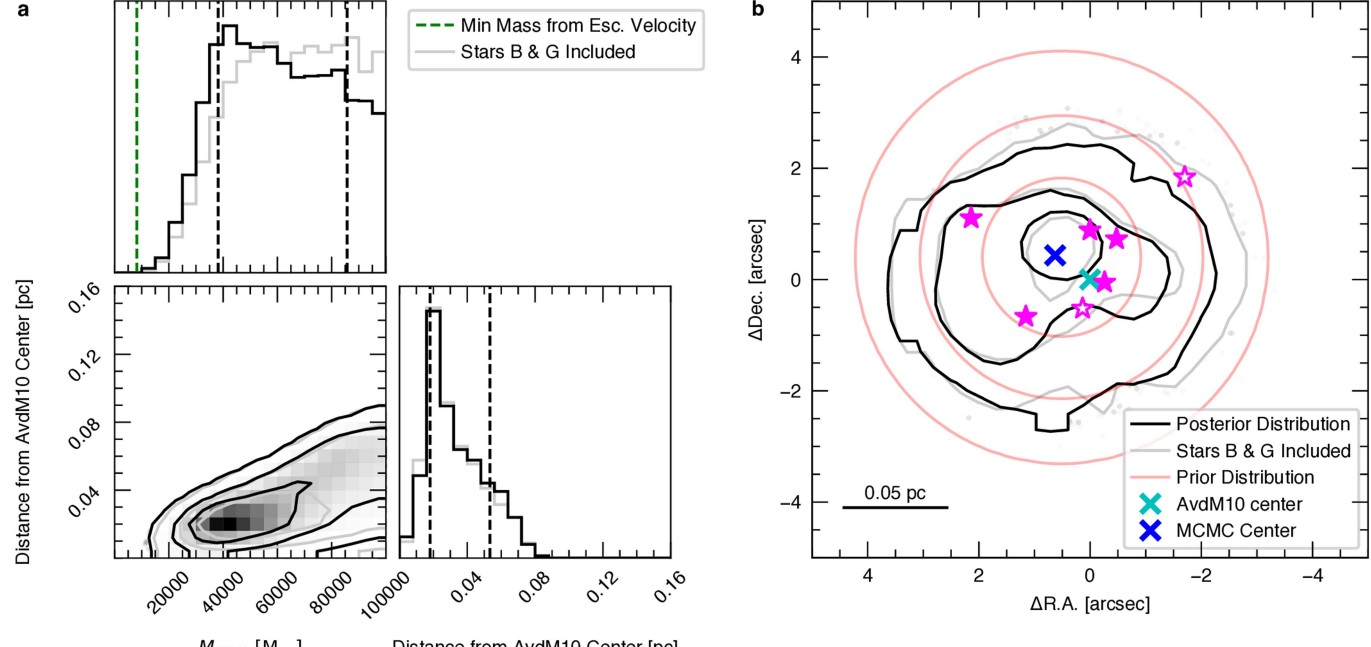

**Extended Data Fig. 6 | Constraints on the IMBH using the acceleration measurements.** Taking into account the limits on the accelerations gives us additional constraints on black hole mass (**a**) and on-sky position (**b**). The contours shown correspond to the 1-, 2-, and 3-sigma levels of the distribution.

This analysis using both escape velocity and acceleration measurements from the five robustly measured fast stars constrains the minimum IMBH mass stronger than escape velocity constraints alone. Both plots also show the distribution from an MCMC run including stars B and G.

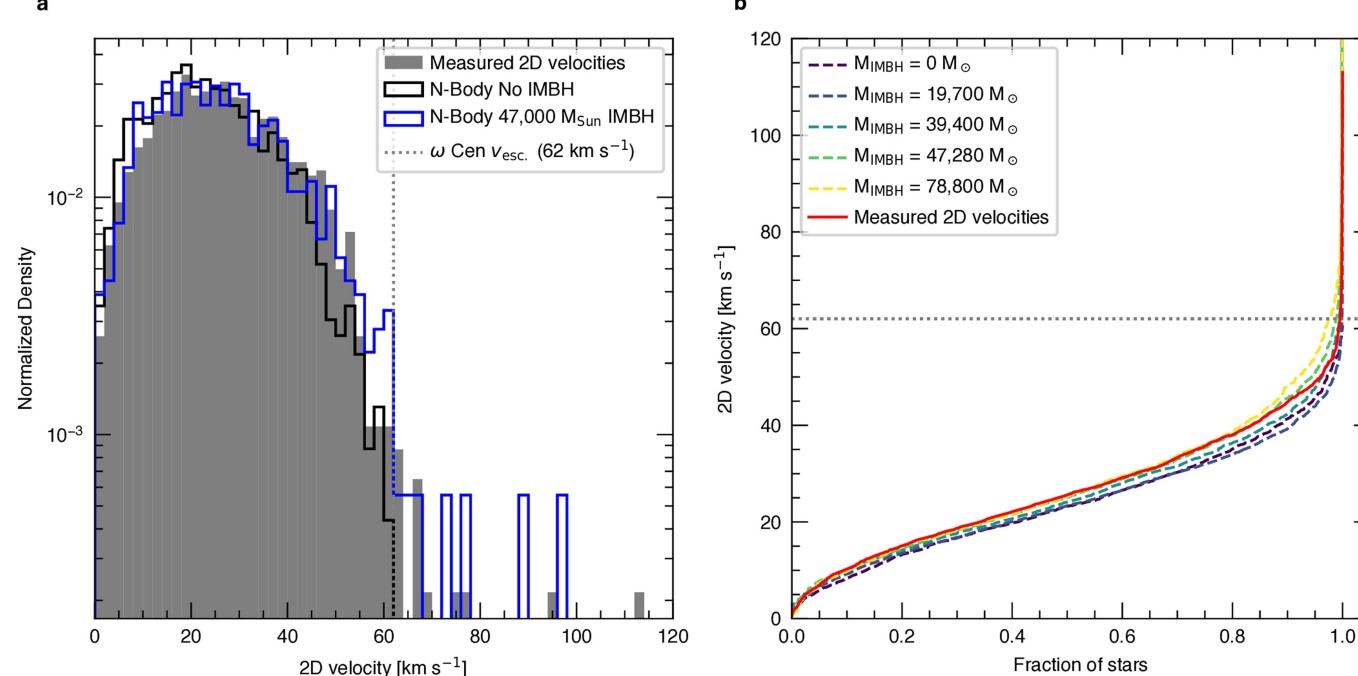

**Extended Data Fig. 7 | Comparison of the observed velocity distribution with N-Body models. a**, 2D velocity distribution for the stars in the inner 10 arcseconds of $\omega$ Cen. We show the observed data in grey, the results for an N-body model without an IMBH in black, and the results for a model with an 47,000 M$_\odot$ IMBH in blue (based on the models of ref. 9). The N-body model without an IMBH predicts no stars above the escape velocity; the 47,000 M$_\odot$ model predicts a number close to our observations. **b**, Comparison of the normalized, cumulative distribution of stellar velocities for our data and five different N-body models.

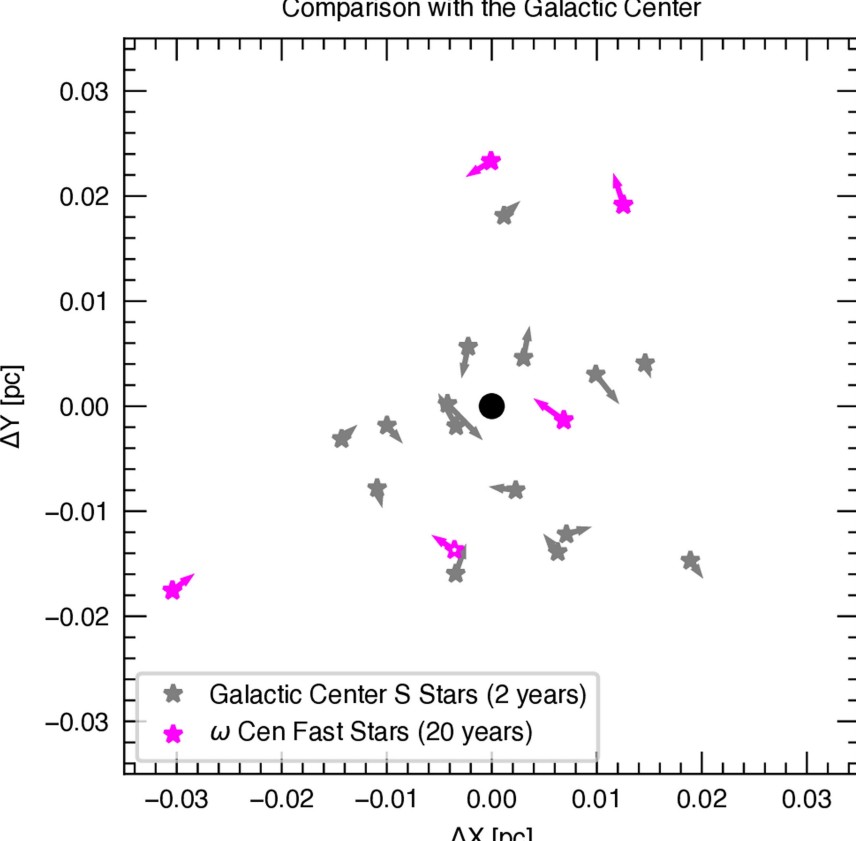

**Extended Data Fig. 8 | Comparison with the Galactic Centre.** In this figure we compare the observed physical motion of our fast star sample with the stars orbiting the black hole Sgr A* in the Galactic Center[17]. The physical scale probed by the fast-moving stars is similar to that probed by the S stars in the Milky Way centre; however, the density of these tracers is lower. Owing to the approximately ~100 times higher black hole mass of Sgr A*, we expect the motions to be ~10 times faster and periods of the stars to be ~10 times shorter, and thus show the motion for 2 years for the S stars to compare to the 20 year time span we observe the stars in ω Cen.

**Extended Data Table 1 | Detailed astrometric and photometric information of the fast-moving stars**

**a**

| | General Properties | | | | |
|---|---|---|---|---|---|
| Star | Catalog ID | R.A. [degree] | Dec. [degree] | $r_{proj.}$ ["] | $N_{used}$ |
| A | 532867 | 201.6967263 | -47.4795835 | 0.265 | 308 |
| B | 475236 | 201.6968888 | -47.4797138 | 0.537 | 353 |
| C | 1422379 | 201.6966378 | -47.4793672 | 0.870 | 262 |
| D | 476492 | 201.6968346 | -47.4793233 | 0.886 | 195 |
| E | 509644 | 201.6973080 | -47.4797545 | 1.333 | 430 |
| F | 476467 | 201.6977125 | -47.4792625 | 2.408 | 427 |
| G | 510061 | 201.6961340 | -47.4790585 | 2.506 | 338 |

**b**

| | Velocity and Acceleration Measurements | | | | | |
|---|---|---|---|---|---|---|
| Star | PM R.A. $\mu_\alpha\cos\delta$ [mas yr$^{-1}$] | PM Dec. $\mu_\delta$ [mas yr$^{-1}$] | Total PM [mas yr$^{-1}$] | $v_{2D}$ [km s$^{-1}$] | Acceleration R.A. [mas yr$^{-2}$] | Acceleration Dec. [mas yr$^{-2}$] |
| A | 3.563±0.038 | 2.564±0.055 | 4.390±0.044 | 113.0±1.1 | -0.0069±0.0083 (0.8σ) | 0.0085±0.0098 (0.9σ) |
| B | 2.167±0.182 | 1.415±0.081 | 2.588±0.159 | 66.6±4.1 | 0.0702±0.0239 (2.9σ) | 0.0228±0.0157 (1.5σ) |
| C | 1.117±0.127 | 3.514±0.056 | 3.687±0.066 | 94.9±1.7 | 0.0028±0.0333 (0.1σ) | -0.0060±0.0123 (0.5σ) |
| D | 2.559±0.082 | -1.617±0.061 | 3.027±0.076 | 77.9±2.0 | 0.0357±0.0177 (2.0σ) | -0.0194±0.0162 (1.2σ) |
| E | -2.149±0.025 | 1.638±0.037 | 2.702±0.030 | 69.6±0.8 | 0.0072±0.0042 (1.7σ) | -0.0009±0.0075 (0.1σ) |
| F | 0.436±0.017 | -2.584±0.016 | 2.620±0.016 | 67.4±0.4 | 0.0052±0.0038 (1.4σ) | -0.0015±0.0038 (0.4σ) |
| G | -1.317±0.098 | 2.207±0.062 | 2.571±0.073 | 66.2±1.9 | -0.0197±0.0267 (0.7σ) | 0.0173±0.0170 (1.0σ) |

**c**

| | Photometric Properties | | | | | | | |
|---|---|---|---|---|---|---|---|---|
| Star | $m_{F606W}$ | $QFIT_{F606W}$ | $RADX_{F606W}$ | $f_N/f._{F606W}$ | $m_{F814W}$ | $QFIT_{F814W}$ | $RADX_{F814W}$ | $f_N/f._{F814W}$ |
| A | 23.373±0.009 | 0.976 (p =0.716) | -0.006 (p =0.192) | 0.366 (p =0.532) | 22.134±0.027 | 0.945 (p =0.170) | -0.004 (p =0.249) | 0.363 (p =0.537) |
| B | 22.778±0.014 | 0.940 (p =0.096) | 0.082 (p =0.981) | 0.122 (p =0.322) | 21.625±0.026 | 0.956 (p =0.128) | 0.072 (p =0.964) | 0.494 (p =0.639) |
| C | 23.630±0.009 | 0.940 (p =0.283) | 0.050 (p =0.8249) | 0.147 (p =0.320) | 22.342±0.034 | 0.916 (p =0.109) | -0.014 (p =0.136) | 0.327 (p =0.514) |
| D | 24.108±0.017 | 0.952 (p =0.585) | -0.008 (p =0.216) | 0.064 (p =0.180) | 22.745±0.037 | 0.952 (p =0.291) | -0.056 (p =0.022) | 0.079 (p =0.198) |
| E | 20.112±0.004 | 0.996 (p =0.131) | 0.014 (p =0.950) | 0.424 (p =0.923) | 19.373±0.009 | 0.995 (p =0.109) | 0.018 (p =0.938) | 0.622 (p =0.975) |
| F | 21.348±0.004 | 0.997 (p =0.635) | -0.002 (p =0.340) | 0.028 (p =0.253) | 20.391±0.004 | 0.998 (p =0.799) | 0.004 (p =0.587) | 0.032 (p =0.232) |
| G | 20.888±0.009 | 0.980 (p =0.031) | -0.003 (p =0.219) | 2.049 (p =0.997) | 20.069±0.014 | 0.983 (p =0.053) | 0.007 (p =0.730) | 0.971 (p =0.984) |

The table in **a** shows the position, the number of astrometric measurements $N_{used}$, and the projected distance from the AvdM10 centre for each of the fast-moving stars. The table in **b** lists the individual proper motion components, the total proper motion, the inferred 2D velocity and the measurements of the acceleration for the seven fast-moving stars. All shown errors correspond to the 1σ errors, which were estimated by scaling the formal errors on the parameters by $\sqrt{\chi^2_{red.}}$ of the respective fit. The strongest accelerations are shown by Star B; however, this star has been discarded from the set of robustly measured stars because of large astrometric errors. All robust stars show an acceleration consistent with zero to within 2σ. Finally, the table in **c** lists several photometric properties of the fast-moving stars in two filters, including the measured brightness and several photometric diagnostics (the quality of fit (QFIT) parameter, the radial excess (RADX) parameter and the flux ratio between the flux of each source and the flux of neighbouring sources). Together with the photometric diagnostic values we show their percentile with respect to stars with a similar magnitude (*p*).

**Extended Data Table 2 | Comparison of the observed velocity distribution with N-body models**

| IMBH mass fraction in model | $M_{IMBH}$ [$M_\odot$] | $N_{10''total}$ | KS test p-Value | $N_{3'',fast}$ | $N_{3'',total}$ | $N_{3'',fast}/N_{3''total}$ |
|---|---|---|---|---|---|---|
| Observed | Data | 2,324 | - | 7 | 218 | 3.2% |
| 0 | 0 | 1,148 | 6.21e-7 | 0 | 113 | 0% |
| 0.5% | 20,000 | 1,342 | 8.60e-10 | 0 | 113 | 0% |
| 1.0% | 39,000 | 868 | 0.0027 | 3 | 89 | 3.4% |
| 1.2% | 47,000 | 901 | 0.27 | 7 | 85 | 8.3% |
| 2.0% | 78,800 | 843 | 0.18 | 7 | 70 | 10.0% |

In this table we report results of the comparison of the observed distribution of 2D velocities with different N-body models (see also Extended Data Fig. 8). The first two columns indicate the relative and absolute mass of the IMBH in the N-body models. The third column shows the total number of stars within 10″ of the cluster centre. In the fourth column, we show the results of a Kolmogorov-Smirnov test comparing the measured velocity distribution with the different N-body models. Finally, the last three columns compare the absolute number and the fraction of fast-moving stars in the centremost 3 arcseconds.