## [Peer Review File · Nature]

Manuscript Title: Fast-moving stars around an intermediate-mass black hole in ω Centauri

Reviewer Comments & Author Rebuttals

Reviewer Reports on the Initial Version:

Referees' comments:

Referee #1 (Remarks to the Author):

This is an important, exciting paper proving the discovery of a bona fide intermediate-mass black hole as the central black hole in the stripped galaxy nucleus Omega Cen, only ~ 5 kpc away. This is a landmark built on careful observational analysis that will garner vast numbers of citations and secure careers. All of the essential analysis is well-done. I have a number of suggestions the authors should carefully consider before I can recommend the paper for publication, but these are mostly improvements or clarifications to how the results are presented; none touch on the fundamentals.

--The text on pg 5 needs to mention the discrepancy revealed by the consistency check of the N-body model with the relative # of fast and slow stars revealed in Table 4/pg 16, which could reflect the need for improved modeling of stellar-mass black holes or another issues. While it does not challenge the existence of the intermediate-mass black hole, it suggests substantial uncertainty on the precise mass.

--The main text of the paper would benefit from a clarifying sentence somewhere discussing the relationship between Omega Cen and other Milky Way globular clusters. I expect this paper to set off a "gold rush" of people attempting to once again find intermediate-mass black holes in globular clusters. While this present paper is very clear regarding the compelling evidence that Omega Cen is a stripped galactic nucleus, I think it would be worth explicitly stating that while this means it's a really good idea to spend effort looking for an intermediate-mass black hole in the other likely stripped nucleus (M54), and to a corresponding lesser degree in the clusters of less certain provenance (NGC 2419?), it says very little about whether intermediate-mass black hole

are more likely to occur in "normal" globular clusters that are not tidally stripped galaxy nuclei, except perhaps to suggest one may need to examine fainter stars for PMs than previous expectations, and to take less seriously stated limits on the basis of a lack of fast stars (as was done for decade(s) with Omega Cen).

--The discussion of the implications of the lack of detection in X-ray and radio should be clarified. For example, pg 3, 90-92: "the very deep upper limits on a central source in both X-ray [22] and radio [23] wavelengths suggest either no intermediate-mass black hole is present, or that the gas density is even lower than what is typically observed in globular clusters."

Conditional on the existence of a black hole, X-ray observations constrain a combination of the mass accretion rate and radiative efficiency, and in the absence of other information separating these components is not possible. A low gas density could be one reason for a low accretion rate, but there could be others; also, the gas density could be as expected, but the accretion could have lower radiative efficiency than assumed. Radio observations add an additional complication in that they are linked to the accretion flow by the uncertain fundamental plane, adding another poorly-understood physical parameter. The text in the paper should be corrected appropriately. The similar statement on pg 6, 204-206, which says that "This is fainter than expected based on the typical gas density observed in globular clusters from pulsar dispersion measurements" is also not quite correct --- the relevant uncertain quantities include all of the gas density, the accretion rate, and the radiative efficiency. The next sentence is accurate, so this should be able to easily be rewritten.

--In Figure 3, the faintest 4 "fast" stars all appear to be on the blue side of the main sequence to an unusual degree---perhaps suggesting they are all more metal-poor than average? I'm not sure what the color-magnitude diagram of the central stars alone looks like, which could also be relevant. It might be worth mentioning this in the context of the discussion of how the bound stars do not appear to be uniformly selected/representative of the Omega Cen stars.

--The discussion of acceleration limits on pg 12 seems to gloss over how useful these are (basically saying they are not very useful). While the listed example is true enough, I feel like it undersells

the parts of parameter space that are well-constrained. For example, if the exact Anderson10 center is used, then star A seems quite constraining: it has a projected distance of only 0.27", which would imply an acceleration of ~ 3.5 km/s/yr (0.14 mas/yr/yr) if this was its true distance and the intermediate-mass black hole was 40000 Msun. But this predicted acceleration is very strongly ruled out by its upper limit, suggesting either a lower mass or a greater distance. Sure, star A could be much further away in three dimensions even if the projected distance is the same, but the fact that it has the highest proper motion velocity of all the stars in the sample (by a lot!) suggests this is unlikely. The analysis on pg 15 implies that a high mass *is favored* despite the low acceleration of star A, which the model seems to solve by moving the intermediate-mass black hole away from star A so it's no longer very close.

That's all a bit discursive---my point is that I think the accelerations *do* matter quite a lot for the pg 15 modeling, which isn't the impression one gets from pg 12. I think this could be clarified, potentially also with information about which stars end up constraining the location + black hole mass the most. For example, star D seems to end up close to the favored intermediate-mass black hole location, perhaps because of its acceleration.

--Also, while Figure 10 is nice, I'd like to see some version of its right panel which has the most likely location of the black hole shown with respect to the Anderson10 center *in the context of the HST image* for improved visualization. The paper also needs to list the best-fit intermediate-mass black hole location in RA/Dec coordinates including the 1-sigma (or, if they want, 2-sigma) posterior uncertainty in the relevant location on pg 15, not just the (x,y) offsets from the Anderson10 center in pc, and in addition to this in one of the tables (or in a new table) the offsets of the fast 7 from the intermediate-mass black hole peak posterior center, since some of these are quite different from the Anderson10 center offsets.

Finally, I have two smaller issues with the text:

--boldface paragraph: "after the Milky Way center, only the second where we can track the orbits of multiple individual bound companions."

There are of course a number of other massive BHs that have "multiple individual bound companions": those that host maser disks (the masers are multiple, individually observed, bound,

and it seems reasonable to call them companions!) This is already an exciting result, there is no need to stretch to make additional claims about it: you can just call it the nearest massive black hole and stop there. This incorrect statement is repeated on pg 6, 194-196 and on pg 16.

--pg 2, 53-56: "The James Webb Space Telescope has recently revealed a population of rapidly accreting, massive black holes at high redshift, favoring "heavy" seeding scenarios that involve the direct collapse of $\sim 105 M_{\odot}$ black holes in the early universe [10, 11]."

This is not an accurate summary of citation [11], which does not conclude this. [11] notes that while heavy seeds can get one to high masses at high redshift, the observed number density of high-z BHs may actually be difficult to produce in a heavy seed model, and a light seed model that involves super-Eddington growth and/or mergers could potentially better explain the data. The object discussed in citation [10] would indeed likely require a heavy seed, but the actual observed X-ray flux from this object is not very high and the inferred intrinsic luminosity (and hence BH mass) comes entirely from an enormous N_H correction in a low-count spectrum. The authors should reword this statement with a more balanced one.

Referee #2 (Remarks to the Author):

I start with a congratulations to the authors on their excellent work in the data analysis and in the result. It was clearly a lot of hard work, which they detail well. The implications are important and worthy of publication in Nature.

The comments that I provide below do not effect the overall

implications or require re-analysis. In addition to some important scientific considerations, my other comments have to do with the style and strong statements. I understand the move in the community to using strong rhetoric, and suggesting that their work is both the first to be done and the most conclusive. When these statements are made, it is very often neither the first nor the most conclusive. This is true for this paper, which I will outline below.

I am confident modest re-wording of the paper throughout the draft will enhance their nice result.

I list these comments in order of importance:

1) Measure and use of escape velocity

The escape velocity from Baumgardt et al. is based on their dynamical modeling. This paper uses the escape velocity as the primary claim for a black hole, and that escape velocity is from that dynamical model. Yet they claim "these model-independent constraints purely based on our astrometric data" (line 161). Stating that their result is model independent is simply incorrect. In fact, it is less robust in many ways compared to previous models. Basically, you cannot have the result depend on the escape velocity and call it model independent.

In particular, they make a very strong claim in line 214: "In contrast to all previous dynamical (non-)detections, this result does not depend on modeling assumptions". This is wrong, and it would be very hard to argue anything different. I would be interested to hear a rebuttal, and that their result does not require the model-dependent escape velocity.

For example, the models with no black hole from Noyola et al. have an escape velocity around 85 km/s. Baumgardt gets such a lower value of 62. It looks like Baumgardt et al. are using an out-of-date velocity distribution. The very careful analysis from Noyola et al. has a central dispersion of 25 km/s, compared to the value of 16 that Baumgardt is using.

Their other estimates of the escape velocity based on the surface brightness profile are significantly less secure. They also depend on the tidal radius, which is highly uncertain. I would not have included these estimates since they don't add too much. But again, they are very model dependent.

The authors need to correct this confusion throughout the paper. I.e., their claim of a black hole is very model dependent, opposite to what is currently stated.

2) The authors should have a more complete discussion of using

individual velocity estimates and a modeled escape velocity to infer the enclosed mass. The first paper to attempt this is Gunn & Griffin (1979), and I was quite surprised to not see a reference to this important paper. Another important one is Meylan, Dubath, Mayor (1991, ApJ, 383). In both of these papers, it was quickly realized how hard it is to infer enclosed mass from the escape velocity arguments. The Meylan result did not last since the escape velocity was re-measured to be higher. I have some concern of the same in this paper. Can the authors address these issues?

I suggest a paragraph on both the pitfalls and the advantages of using individual velocities in the tail of the distribution combined with the escape velocity uncertainty.

Basically, in no way would I call this result "conclusive". That word implies there is no other interpretation. There is no loss of importance for the paper if they remove "Conclusive" from the title, and I strongly suggest that.

3) Comparison to Noyola et al.: The authors are fairly quick to dismiss the result from Noyola et al. based on arguments not addressed. First, the authors should consider the best fit mass and uncertainties from Noyola et al. These are $4.7(+1)e4$ Msun. And this should be compared to their value of $6(+2.4)e4$ Msun. I ask that the authors present these sets of numbers in the same paragraph somewhere.

4) The uncertainties on the BH mass from Noyola are over 2x smaller than those from this work. If we all trust each others statistics, this would imply that the Noyola result is more conclusive. I believe the take of the authors is that they do not trust the Noyola values. If so, they need to state that the Noyola result, while more accurate, is not as reliable.

I argue strongly that the dynamical model being used by Noyola et al. is significantly more robust and more general than the dynamical model being used by the authors. The orbit-based modeling in Noyola is state-of-the-art (whatever that means), in terms of providing the most general orbital structure. The dynamical models being used by the authors is based on n-body simulations that have a very large number of buried assumptions. The authors, in line 437 on N-body simulations, mention just a few parameters. There are others not being considered like assumption of King Model, initial conditions, MW tidal effects, IMF, initial-to-final mass for white dwarf, neutron star population, natal kick distribution of ns and stellar-mass BH. And many others. The idea that they can capture all of these and think they can provide a solid measure of the escape velocity is not correct. The orbit-based models used by Noyola et al. take the point that we simply use all available phase space to provide the most general constraints on the BH mass.

Thus, many in the community would argue the dynamical models in Noyola et al. are more robust than the N-body models used here. Furthermore, this is reflected in the better uncertainties in Noyola et al. The authors should address these issues, and at the least back off on the quick dismissal of work that some consider stronger.

5) The histogram of the 2d velocities in Fig 11 shows a value at 109 km/s that is not considered in Table 1. I checked the others and those seem to be fine, but the authors need to double check. I would normally just treat this as some entry mistake, but since it would be the 2nd highest velocity, I ask the authors explain what happened here. I suspect it might have been excluded due to their cuts, but just didn't make it out of the figure generation. If there are points with high velocity not present, it might be important to discuss in the paper. I leave that to the authors.

6) Are the high velocities really that extreme? Noyola et al present a velocity dispersion of 25 km/s. The n-sigmas for each of the 7 stars, assuming a normal distribution, are then 4.5, 2.7, 3.8, 3.1, 2.8, 2.7, 2.6. Of course, it depends on the exact shape, especially in the tails, and also the total number of proper motions, but the these numbers seem to be consistent with expectations. I'm guessing the major issue is that Noyola measures 25 km/s, and Baumgardt measure 16 km/s. Can the authors make it clear what is going on here?

7) line 66: G1, from Gebhardt, Rich, Ho (2002 and 2005) still stands out as one of the best measure BHs in this range. At the least, the authors should cite these papers. And I suggest to try not to dismiss the work when citing.

8) The whole comparison to the MWBH and the S-stars is out of place. I'm not sure of the point and it takes away from the paper. If the point is to explain why they don't see accelerations, then that is already explained well. I suggest removing this section and plot.

9) These sections and figure are very well done. I only comment:

Fig 4 and Fig 6 are great. It is very important and useful to be included. But it is also very scary for many of the fast-moving stars. It demonstrates the careful work of the authors. I thank the authors for including these.

The concern of background/foreground contamination is well considered. It is convincing that contamination is not a problem.

10) fig 10 caption: "robuseely"

REVIEWER: Karl Gebhardt

Referee #3 (Remarks to the Author):

Conclusive evidence for an intermediate-mass black hole in ω Centauri

by Häberle et al.

Nature manuscript 2023-12-22619

Summary

This manuscript presents the discovery of 7 faint, potentially fast-moving stars in the direction of ω Cen (really 5 stars in the robust sample). The large stellar velocities are derived from high precision PM measurements over a 20-year baseline, assuming the stars are at the cluster distance. To validate this assumption, the authors present evidence that these stars are cluster members (rather than Galactic foreground/ background stars or other contaminants), e.g., they associate the stars with the CMD of the cluster and place them within the inner 3 arcsec of the cluster centre. With this set of measurements and assumptions, they argue that an intermediate mass black hole at the centre of ω Cen is the most likely explanation for the over density and high velocities of these unusual stars.

This is a potentially exciting result that is deserving of publication in Nature, but the evidence for the main claim of the paper (discovery of an IMBH in ω Cen) is overstated. For example, the abstract claims these results “settle a two-decade-long debate about the existence of an IMBH in ω Cen and provide the first robust black hole detection in the intermediate-mass regime” (lines 45-47). This statement is too strong given uncertainties in the analysis, as outlined by the authors and detailed below. The results are tantalizing and the associated PM catalog is truly exceptional, so I believe this result could be published in Nature but it requires a shift in the language, e.g., “new evidence for the presence of a black hole”. Along these same lines, the title of the paper overstates the strength of the findings — these are beautiful measurements that might support the existence of an IMBH in ω Cen, but the evidence is not “conclusive” (see also line 187).

Major Comments

I have several major concerns about the conclusions drawn from the measurements presented in the paper:

1. The authors argue that their sample of high velocity stars is associated with the cluster and must be bound to an IMBH to explain their high velocities. However, they do not have a firm constraint on the stellar orbits either from line-of-sight velocities or astrometric deviations/accelerations. They present a discussion of accelerations measured via astrometry (lines 312 to 329), but cannot place constraints on the majority of their sample except for two stars: Star B (rejected for most of the analysis) and Star D which is the faintest star in the high-velocity sample. Hence, the authors consider the measures from these two stars unreliable.

Meanwhile, spectroscopic, line-of-sight velocities are mentioned in lines 217-223 and lines 343-360. In particular, Stars E and F have LOS velocities from the VLT/MUSE sample “oMEGACat” which appear consistent with the systematic LOS of wCen, but these stars are farther from the centre and don’t offer conclusive evidence for an IMBH on their own. To make this a conclusive/robust measure of a globular cluster IMBH, one of these avenues must be pursued. For example, targeted observations to obtain LOS velocities for the other 5 stars in the sample would greatly strengthen the case for an IMBH.

2. The authors emphasize that 3 of the innermost and fastest moving stars are also the faintest in the sample (Stars A, C, and D; lines 131 to 141). They use this to hint at a possible ejection/capture scenario for the existence of a population of low mass, very high velocity stars. However, the faint magnitudes of these stars also imply that they are the most likely to have large uncertainties in their astrometry and photometry. There is considerable discussion in the paper about the astrometric uncertainties, but it’s not clear how robustly these stars can be associated with the main sequence of wCen, which also calls their cluster membership into question. The authors need to discuss this in more detail since cluster membership is fundamental to all of their analysis.

3. The discussion of high velocities associated with ejection from triple systems is insufficient (lines 177-185). The authors must be more specific about how many additional high velocity stars would be expected at larger radii if ejections are common. The authors also claim the “high rate” of fast moving stars implies that the cluster would be depleted rapidly if ejection were common, but they do not clarify what rate they are using, e.g., is it based on the 5-star sample or the 7-star

sample? What if all but 2 of these stars are contaminants? This scenario requires more detailed treatment since dynamical interactions are well-known to occur in massive stellar clusters like wCen.

4. In lines 197 to 210, the authors mention that other MWL constraints taken together with their results imply an extremely low Eddington ratio for their claimed IMBH. They even point out that newly discovered pulsars in wCen might offer a direct measure of the gas density in the cluster, but they do not pursue this possibility. The authors should make at least a rough estimate of the gas density based on the pulsar DMs (if they are published) and check whether it is consistent with the low Eddington ratio they require.

Minor Comments

I have a series of other, more minor comments on the analysis, text, figures and tables:

1. In line 232 and lines 236-237, the authors call the PM catalog “high-precision” and then claim “unprecedented depth and precision”. These statements need to be quantified and ideally associated with a literature reference.

2. The acronym MGE is never defined (first used line 371), I assume this is “multi-Gaussian expansion” but the authors should clarify and explain (briefly) why these are the appropriate models to use.

3. Lines 435-436 — it’s not clear what the “delta” is in reference to; is this relative to the Anderson’10 cluster centre?

4. Figure 1, left panel: the magenta on blue in this figure is difficult to make out. Please consider making the blue symbols light grey to improve readability. The proper motion vectors in the right-hand panel are also hard to make out; perhaps the authors could remove the legend on the right-hand-side (since it is the same in both panels) and zoom in a bit further.

5. Figure 3 — no error bars are given for the colour-magnitude diagram so it's not clear how confidently associated the fast-moving stars are with the cluster main-sequence, particularly at the faint end; perhaps a "characteristic" set of error bars for the brightest and faint target could be added for reference.

6. Similar comment for Fig 5 — it's not clear how close or far the targets of interest are from the primary sequences of the cluster stars since there are no error bars.

7. In Figure 6, Stars B, E, and G appear to suffer from confusion. Stars B and G are often omitted in the analysis, but Star E is not similarly flagged. Why keep Stars B and G in the analysis at all? They muddy the narrative and even the possible acceleration measurement for Star B does not improve the fit. I would like to see a concrete argument for including Stars B and G, otherwise I suggest removing them from the sample.

8. Figure 7 — the two blue histograms are very similar and hard to tell apart. Also, it looks like the distribution in magnitude is bimodal. If all of the fast-moving stars are in one part of the bimodal distribution, might this impact their claimed association with the CMD and thus their cluster membership?

9. There are two typos in the caption for Figure 9: In the second sentence, it should be "profiles" and in the 3rd-to-last line it should be "pink markers".

Author Rebuttals to Initial Comments:

Referees' comments:

Referee #1 (Remarks to the Author):

This is an important, exciting paper proving the discovery of a bona fide intermediate-mass black hole as the central black hole in the stripped galaxy nucleus Omega Cen, only ~ 5 kpc away. This is a landmark built on careful observational analysis that will garner vast numbers of citations and secure careers. All of the essential analysis is well-done. I have a number of suggestions the authors should carefully consider before I can recommend the paper for publication, but these are mostly improvements or clarifications to how the results are presented; none touch on the fundamentals.

R1-1 --The text on pg 5 needs to mention the discrepancy revealed by the consistency check of the N-body model with the relative # of fast and slow stars revealed in Table 4/pg 16, which could reflect the need for improved modeling of stellar-mass black holes or another issues. While it does not challenge the existence of the intermediate-mass black hole, it suggests substantial uncertainty on the precise mass.

We agree that this tension shows the limits of the existing N-Body models of w Cen and has to be accounted for in future modeling efforts. We have added a comment about this in the main part of the paper, it was also already mentioned in the section about N-Body models in the Methods section.

R1-2 --The main text of the paper would benefit from a clarifying sentence somewhere discussing the relationship between Omega Cen and other Milky Way globular clusters. I expect this paper to set off a "gold rush" of people attempting to once again find intermediate-mass black holes in globular clusters. While this present paper is very clear regarding the compelling evidence that Omega Cen is a stripped galactic nucleus, I think it would be worth explicitly stating that while this means it's a really good idea to spend effort looking for an intermediate-mass black hole in the other likely stripped nucleus (M54), and to a corresponding lesser degree in the clusters of less certain provenance (NGC 2419?), it says very little about whether intermediate-mass black hole are more likely to occur in "normal" globular clusters that are not tidally stripped galaxy nuclei, except perhaps to suggest one may need to examine fainter stars for PMs than previous expectations, and to take less seriously stated limits on the basis of a lack of fast stars (as was done for decade(s) with Omega Cen).

We have added a discussion about other potential stripped nuclei at the end of the main part of the paper.

R1-3 --The discussion of the implications of the lack of detection in X-ray and radio should be clarified. for example, pg 3, 90-92: "the very deep upper limits on a central source in both X-ray [22] and radio [23] wavelengths suggest either no intermediate-mass black hole is present, or that the gas density is even lower than what is typically observed in globular clusters."

Conditional on the existence of a black hole, X-ray observations constrain a combination of the mass accretion rate and radiative efficiency, and in the absence of other information separating these components is not possible. A low gas density could be one reason for a low accretion rate, but there could be others; also, the gas density could be as expected, but the accretion could have lower radiative efficiency than assumed. Radio observations add an additional complication in that they are linked to the accretion flow by the uncertain fundamental plane, adding another poorly-understood physical parameter. The text in the paper should be corrected appropriate. The similar statement on pg 6, 204-206, which says that "This is fainter than expected based on the typical gas density observed in globular clusters from pulsar dispersion measurements" is also not quite correct --- the relevant uncertain quantities include all of the gas density, the accretion rate, and the radiative efficiency. The next sentence is accurate, so this should be able to easily be rewritten.

We have revised the relevant text and moved it into the Methods section under "Previous Accretion Constraints in Context"

R1-4 --In Figure 3, the faintest 4 "fast" stars all appear to be on the blue side of the main sequence to an unusual degree---perhaps suggesting they are all more metal-poor than average? I'm not sure what the color-magnitude diagram of the central stars alone looks like, which could also be relevant. It might be worth mentioning this in the context of the discussion of how the bound stars do not appear to be uniformly selected/representative of the Omega Cen stars.

We have verified that this is not a systematic effect caused e.g. by local differential reddening.

Indeed this is quite intriguing. Although numbers are quite low, this could contain some information on the capture mechanism for these stars (together with the fact that all of them are quite faint).

We now discuss this in the main part of the manuscript.

R1-5 --The discussion of acceleration limits on pg 12 seems to gloss over how useful these are (basically saying they are not very useful). While the listed example is true enough, I feel like it undersells the parts of parameter space that are well-constrained. For example, if the exact Anderson10 center is used, then star A seems quite constraining: it has a projected distance of only 0.27", which would imply an acceleration of ~ 3.5 km/s/yr (0.14 mas/yr/yr) if this was its true distance and the intermediate-mass black hole was 40000 Msun. But this predicted acceleration is very strongly ruled out by its upper limit, suggesting either a lower mass or a greater distance. Sure, star A could be much further away in three dimensions even if the projected distance is the same, but the fact that it has the highest proper motion velocity of all the stars in the sample (by a lot!) suggests this is unlikely. The analysis on pg 15 implies that a high mass *is favored* despite the low acceleration of star A, which the

model seems to solve by moving the intermediate-mass black hole away from star A so it's no longer very close.

That's all a bit discursive---my point is that I think the accelerations *do* matter quite a lot for the pg 15 modeling, which isn't the impression one gets from pg 12. I think this could be clarified, potentially also with information about which stars end up constraining the location + black hole mass the most. For example, star D seems to end up close to the favored intermediate-mass black hole location, perhaps because of its acceleration.

We have extended the discussion of the acceleration constraints and their influence on the MCMC run, we also discuss the role of star D which is adding the strongest constraints on the location in the MCMC runs.

R1-6 --Also, while Figure 10 is nice, I'd like to see some version of its right panel which has the most likely location of the black hole shown with respect to the Anderson10 center *in the context of the HST image* for improved visualization. The paper also needs to list the best-fit intermediate-mass black hole location in RA/Dec coordinates including the 1-sigma (or, if they want, 2-sigma) posterior uncertainty in the relevant location on pg 15, not just the (x,y) offsets from the Anderson10 center in pc, and in addition to this in one of the tables (or in a new table) the offsets of the fast 7 from the intermediate-mass black hole peak posterior center, since some of these are quite different from the Anderson10 center offsets.

We now list RA and Dec of the IMBH location in the text. We have also added the best fit IMBH location to Figure 1, which contains the stacked HST image. We think adding the HST stack as background to Figure 10 will make it a bit crowded.

Finally, I have two smaller issues with the text:

R1-7 --boldface paragraph: "after the Milky Way center, only the second where we can track the orbits of multiple individual bound companions."

There are of course a number of other massive BHs that have "multiple individual bound companions": those that host maser disks (the masers are multiple, individually observed, bound, and it seems reasonable to call them companions!) This is already an exciting result, there is no need to stretch to make additional claims about it: you can just call it the nearest massive black hole and stop there. This incorrect statement is repeated on pg 6, 194-196 and on pg 16.

The statement has been removed from the summary paragraph. At the end of the main part we have specified it more clearly: the black hole in wCen is the second, for which we can track the orbit of multiple bound stars.

R1-8 --pg 2, 53-56: "The James Webb Space Telescope has recently revealed a population of rapidly accreting, massive black holes at high redshift, favoring "heavy" seeding scenarios that involve the direct collapse of $\sim 10^5 M_{\odot}$ black holes in the early universe [10, 11]."

This is not an accurate summary of citation [11], which does not conclude this. [11] notes that while heavy seeds can get one to high masses at high redshift, the observed number density

of high- z BHs may actually be difficult to produce in a heavy seed model, and a light seed model that involves super-Eddington growth and/or mergers could potentially better explain the data. The object discussed in citation [10] would indeed likely require a heavy seed, but the actual observed X-ray flux from this object is not very high and the inferred intrinsic luminosity (and hence BH mass) comes entirely from an enormous N_H correction in a low-count spectrum. The authors should reword this statement with a more balanced one.

We thank the referee for these clarifications about the conclusions from the recent Natarajan and Greene papers. Due to editorial concerns about the length of the paper, this whole paragraph has been removed from the manuscript.

Referee #2 (Remarks to the Author):

I start with a congratulations to the authors on their excellent work in the data analysis and in the result. It was clearly a lot of hard work, which they detail well. The implications are important and worthy of publication in Nature.

The comments that I provide below do not effect the overall implications or require re-analysis. In addition to some important scientific considerations, my other comments have to do with the style and strong statements. I understand the move in the community to using strong rhetoric, and suggesting that their work is both the first to be done and the most conclusive. When these statements are made, it is very often neither the first nor the most conclusive. This is true for this paper, which I will outline below.

I am confident modest re-wording of the paper throughout the draft will enhance their nice result.

I list these comments in order of importance:

R2-1

1) Measure and use of escape velocity

The escape velocity from Baumgardt et al. is based on their dynamical modeling. This paper uses the escape velocity as the primary claim for a black hole, and that escape velocity is from that dynamical model. Yet they claim "these model-independent constraints purely based on our astrometric data" (line 161). Stating that their result is model independent is simply incorrect. In fact, it is less robust in many ways compared to previous models. Basically, you cannot have the result depend on the escape velocity and call it model independent.

In particular, they make a very strong claim in line 214: "In contrast to all previous dynamical (non-)detections, this result does not depend on modeling assumptions". This is wrong, and it would be very hard to argue anything different. I would be interested to hear a rebuttal, and that their result does not require the model-dependent escape velocity.

For example, the models with no black hole from Noyola et al. have an escape velocity around 85 km/s. Baumgardt gets such a lower value of 62. It looks like Baumgardt et al. are using an out-of-date velocity distribution. The very careful analysis from Noyola et al. has a central dispersion of 25 km/s, compared to the value of 16 that Baumgardt is

using.

Their other estimates of the escape velocity based on the surface brightness profile are significantly less secure. They also depend on the tidal radius, which is highly uncertain. I would not have included these estimates since they don't add too much. But again, they are very model dependent.

The authors need to correct this confusion throughout the paper. I.e., their claim of a black hole is very model dependent, opposite to what is currently stated.

We agree that assuming a specific escape velocity makes our result not truly model independent - we thank the referee for pointing that out and have changed the wording in the respective parts of the paper.

However, the central escape velocity is a quantity that can be determined quite reliably. It also does not depend strongly on the assumed mass distribution in the innermost region or the retention of remnants, but rather on the total assumed cluster mass, mass-to-light ratio and the assumed distance. Due to new measurements with Gaia these parameters can be determined quite reliably. To emphasize and illustrate this further, we have significantly extended the "Testing the Robustness of the Assumed Escape Velocity" section in the Methods.

The adopted value of 62 km/s is based on N-Body models that are fit to state-of-the-art datasets including Gaia Proper Motions at larger radii, HST proper motion velocity dispersion measurements and MUSE LOS velocity measurements for thousands of stars. In comparison to the surface brightness based profiles, they also allow for an extended distribution of stellar mass black holes.

To verify the robustness of the escape velocity based on the N-Body models (that provide the adopted 62 km/s limit) we ran additional tests varying the assumed IMF and the retention fraction of black holes, the results of this test are listed in Extended Data Figure 5, none of the N-Body models without an IMBH provides a central escape velocity larger than 64.8 km/s.

Beside these tests of the N-Body models, we study the robustness of the central escape velocity using various literature surface brightness profiles, distances and M/L ratios including the ones used by Noyola+2008 and v.d.Marel&Anderson 2010.

Due to the closer distance of these models, the resulting velocities are lower (~55km/s), however, if we use the same distance to scale our proper motions, the fast star sample remains unchanged. Models with a larger distance (e.g. Zocchi+2019) predict a central escape velocity very close to our adopted value. We also verified the influence of the tidal radius and found it to be small (<2km/s).

In addition all these profiles predict a relatively flat escape velocity in the inner ~50 arcseconds (see Extended Data Figure 5), making the detection of fast-stars only in the central few arcseconds more compelling.

Finally, we also add an empirical confirmation of the adopted escape velocity value in the manuscript by studying the distribution of stellar velocities in the inner region (see Extended Data Figure 6). Only a single star in the region ($3'' < r < 10''$) has 2D velocity higher than 63.8km/s ; consistent with the expected density of Milky Way contaminants.

R2-2

2) The authors should have a more complete discussion of using individual velocity estimates and a modeled escape velocity to infer the enclosed mass. The first paper to attempt this is Gunn & Griffin (1979), and I was quite surprised to not see a reference to this important paper. Another important one is Meylan, Dubath, Mayor (1991, ApJ, 383). In both of these papers, it was quickly realized how hard it is to infer enclosed mass from the escape velocity arguments. The Meylan result did not last since the escape velocity was re-measured

to be higher. I have some concern of the same in this paper. Can the authors address these issues?

I suggest a paragraph on both the pitfalls and the advantages of using individual velocities in the tail of the distribution combined with the escape velocity uncertainty.

Basically, in no way would I call this result "conclusive". That word implies there is no other interpretation. There is no loss of importance for the paper if they remove "Conclusive" from the title, and I strongly suggest that.

We agree that these papers are important as the first to study fast stars in globular clusters, however, the observed situations are not fully comparable: In both Gunn & Griffin 1979 and Meylan+1991 there were only two stars at larger distances with respect to the cluster center. The detected velocities were only marginally above (or even below) the escape velocity. Therefore, the discussion in these papers was rather focussed on understanding which mechanisms can lead to stars with such high velocities and not how they can remain bound in the innermost cluster center.

The fast stars we found in Omega Centauri show a much stronger concentration towards the center, the number of fast stars is higher (at least 5) and the velocities are significantly higher than any reasonable assumption for the escape velocity.

We have changed the Title from "Conclusive evidence..." to the more neutral "New evidence..."

R2-3

3) Comparison to Noyola et al.: The authors are fairly quick to dismiss the result from Noyola et al. based on arguments not addressed. First, the authors should consider the best fit mass and uncertainties from Noyola et al. These are $4.7(+1)e4$ Msun. And this should be compared to their value of $6(+2.4)e4$ Msun. I ask that the authors present these sets of numbers in the same paragraph somewhere.

We have added the numerical results of the Noyola+2010 paper for both the Noyola+10 and the AvdM10 center and extended the discussion of previous works in a new (first) section of the Methods section. The main result of our paper is not a new dynamical model, but the detection of the fast moving stars around the AvdM10 center and the firm lower limit on an IMBH they provide.

The value of $6(+2.4)e4$ Msun is not based on a dynamical model, but purely on the limits on the upper limits on accelerations of the fast moving stars, which is why we do not list it as a main result.

Indeed in the future careful dynamical modeling taking into account all the new available kinematic data (MUSE LOS velocities as in Nitschai+2023 and Pechetti+2024 (including the counter-rotating core they detect), our new Proper Motions, combined with Gaia data at larger radii) will be necessary to obtain the most stringent constraints on the black hole

mass. We have emphasized this need for dynamical modeling to refine the BH mass estimate in the final sentences of the main body of the paper.

R2-4

4) The uncertainties on the BH mass from Noyola are over 2x smaller than those from this work. If we all trust each others statistics, this would imply that the Noyola result is more conclusive. I believe the take of the authors is that they do not trust the Noyola values. If so, they need to state that the Noyola result, while more accurate, is not as reliable.

I argue strongly that the dynamical model being used by Noyola et al. is significantly more robust and more general than the dynamical model being used by the authors. The orbit-based modeling in Noyola is state-of-the-art (whatever that means), in terms of providing the most general orbital structure. The dynamical models being used by the authors is based on n-body simulations that have a very large number of buried assumptions. The authors, in line 437 on N-body simulations, mention just a few parameters. There are others not being considered like assumption of King Model, initial conditions, MW tidal effects, IMF, initial-to-final mass for white dwarf, neutron star population, natal kick distribution of ns and stellar-mass BH. And many others. The idea that they can capture all of these and think they can provide a solid measure of the escape velocity is not correct. The orbit-based models used by Noyola et al. take the point that we simply use all available phase space to provide the most general constraints on the BH mass.

Thus, many in the community would argue the dynamical models in Noyola et al. are more robust than the N-body models used here. Furthermore, this is reflected in the better uncertainties in Noyola et al. The authors should address these issues, and at the least back off on the quick dismissal of work that some consider stronger.

We do not claim that the N-Body models we use for a rather qualitative comparison provide more robust constraints on the IMBH mass than the Noyola et al. 2010 paper. Indeed there are a few uncertainties in the N-Body models given the relatively small particle number with respect to Omega Centauri, however, we have run some additional tests to verify that the escape velocity estimate does not depend strongly on the assumptions in the model.

Two independent papers (Zocchi et al. 2019 and Baumgardt et al. 2019) have shown that dynamical models fit to the dispersion profile cannot easily distinguish between a cluster of stellar mass black holes concentrated within the core radius of the cluster and an IMBH (see also Aros et al. 2020). This ambiguity is removed by the detection of the fast stars, as is the large uncertainty in the position of the center.

R2-5

5) The histogram of the 2d velocities in Fig 11 shows a value at 109 km/s that is not considered in Table 1. I checked the others and those seem to be fine, but the authors need to double check. I would normally just treat this as some entry mistake, but since it would be the 2nd highest velocity, I ask the authors explain what happened here. I suspect it might have been excluded due to their cuts, but just didn't make it out of the figure generation. If there are points with high velocity not present, it might be important to discuss in the paper. I leave that to the authors.

For the comparison of the velocity distribution with the N-Body models we have used a radius of 10 arcseconds, in order to take into account the different radii of influence for different mass black holes.

We have investigated the 109 km/s source that appears in the histogram, and indeed it is a star at a radius of 9.7 arcsecond. Due to its relatively large astrometric errors it has been removed when applying the updated proper quality cuts (see reply to editor and all referees).

R2-6

6) Are the high velocities really that extreme? Noyola et al present a velocity dispersion of 25 km/s. The n-sigmas for each of the 7 stars, assuming a normal distribution, are then 4.5, 2.7, 3.8, 3.1, 2.8, 2.7, 2.6. Of course, it depends on the exact shape, especially in the tails, and also the total number of proper motions, but these numbers seem to be consistent with expectations. I'm guessing the major issue is that Noyola measures 25 km/s, and Baumgardt measure 16 km/s. Can the authors make it clear what is going on here?

We thank the referee for this suggestion and have ran some further tests:

We think neither 16 km/s nor 25 km/s represent the state of the art for measurements of the dispersion at the AdvM10 center in Omega Centauri. Pechetti et al. 2024 measure a constant dispersion of around 20 km/s in the inner 10 arcseconds (which encompasses the various centers discussed in the literature) using LOS velocities of individual stars. This is compatible with the proper motion based findings of AdvM10, and velocity dispersion measurements from Watkins+2015.

Also the Baumgardt N-Body models (with stellar mass black holes but no IMBH) reproduce a central dispersion value of ~20 km/s:

With a velocity dispersion of 20 km/s, the 7 fast stars have n-sigmas of 5.7, 3.5, 4.8, 3.8, 3.3, 3.3, 3.2.

However, all these velocity dispersion measurements refer to the 1D velocity dispersion. As we are searching for the fast moving stars using their 2D proper motion indeed we expect a larger number of stars with larger n-sigma in the tail of the distribution.

In a sample of 217 stars (the total number of stars within $r < 3$ arcsec) we expect from a 2D Maxwell Boltzmann distribution with $\sigma = 20$ km/s:

- ~2.4 stars above 60 km/s (3 sigma)
- ~0.07 stars above 80 km/s (4 sigma)
- ~0.0087 stars above 90 km/s (4.5 sigma)
- ~0.0007 stars above 100 km/s (5 sigma)

The detection of 7 stars above 3 times the velocity dispersion and especially the 4.8 and 5.7 outliers are statistically very improbable.

We think what is even more important is the velocity excess of these stars above the central escape velocity of the cluster and the difference between the high end of the velocity distribution in the two regions compared in Figure 10 (see also answer to the first comment).

R2-7

7) line 66: G1, from Gebhardt, Rich, Ho (2002 and 2005) still stands out as one of the best measure BHs in this range. At the least, the authors should cite these papers. And I suggest to try not to dismiss the work when citing.

We have cited the G1 papers and listed the mass in the "Putting the results into context" section in the main body of the paper.

R2-8

8) The whole comparison to the MWBH and the S-stars is out of place. I'm not sure of the point and it takes away from the paper. If the point is to explain why they don't see accelerations, then that is already explained well. I suggest removing this section and plot.

While this indeed is a very qualitative comparison and not essential for our analysis, we think there is still some value in comparing the observational situation, considering that the S-Stars in the Galactic Center are the only other known case of stars orbiting a massive black hole. We added this figure to illustrate the similarities (e.g. physical distances) and differences (timescales, density) between the two systems.

R2-9

9) These sections and figure are very well done. I only comment:

Fig 4 and Fig 6 are great. It is very important and useful to be included. But it is also very scary for many of the fast-moving stars. It demonstrates the careful work of the authors. I thank the authors for including these.

The concern of background/foreground contamination is well considered. It is convincing that contamination is not a problem.

Thank you!

R2-10

10) fig 10 caption: "robusedly"

Corrected, thank you!

REVIEWER: Karl Gebhardt

Referee #3 (Remarks to the Author):

Conclusive evidence for an intermediate-mass black hole in ω Centauri

by Häberle et al.

Nature manuscript 2023-12-22619

Summary

This manuscript presents the discovery of 7 faint, potentially fast-moving stars in the direction of ω Cen (really 5 stars in the robust sample). The large stellar velocities are derived from high precision PM measurements over a 20-year baseline, assuming the stars are at the cluster distance. To validate this assumption, the authors present evidence that these stars are cluster members (rather than Galactic foreground/ background stars or other contaminants), e.g., they associate the stars with the CMD of the cluster and place them within the inner 3 arcsec of the cluster centre. With this set of measurements and assumptions, they argue that an intermediate mass black hole at the centre of ω Cen is the most likely explanation for the over density and high velocities of these unusual stars.

This is a potentially exciting result that is deserving of publication in Nature, but the evidence for the main claim of the paper (discovery of an IMBH in ω Cen) is overstated. For example, the abstract claims these results “settle a two-decade-long debate about the existence of an IMBH in ω Cen and provide the first robust black hole detection in the intermediate-mass regime” (lines 45-47). This statement is too strong given uncertainties in the analysis, as outlined by the authors and detailed below. The results are tantalizing and the associated PM catalog is truly exceptional, so I believe this result could be published in Nature but it requires a shift in the language, e.g., “new evidence for the presence of a black hole”. Along these same lines, the title of the paper overstates the strength of the findings — these are beautiful measurements that might support the existence of an IMBH in ω Cen, but the evidence is not “conclusive” (see also line 187).

Major Comments

I have several major concerns about the conclusions drawn from the measurements presented in the paper:

R3-1

1. The authors argue that their sample of high velocity stars is associated with the cluster and must be bound to an IMBH to explain their high velocities. However, they do not have a firm constraint on the stellar orbits either from line-of-sight velocities or astrometric deviations/accelerations. They present a discussion of accelerations measured via astrometry (lines 312 to 329), but cannot place constraints on the majority of their sample except for two stars: Star B (rejected for most of the analysis) and Star D which is the faintest star in the high-velocity sample. Hence, the authors consider the measures from these two stars unreliable.

Meanwhile, spectroscopic, line-of-sight velocities are mentioned in lines 217-223 and lines 343-360. In particular, Stars E and F have LOS velocities from the VLT/MUSE sample “oMEGACat” which appear consistent with the systematic LOS of wCen, but these stars are farther from the centre and don’t offer conclusive evidence for an IMBH on their own. To make this a conclusive/robust measure of a globular cluster IMBH, one of these avenues must be pursued. For example, targeted observations to obtain LOS velocities for the other 5 stars in the sample would greatly strengthen the case for an IMBH.

We agree that only 3D velocities (including the line-of-sight direction) and acceleration measurements can fully constrain the orbit of the stars and would allow direct measurements of the mass of the IMBH. Therefore, deep follow-up observations are of high importance and we mention this in the last section of the paper.

However, the 2D velocities inferred from the proper motions are only a lower limit of the 3D velocity, and therefore the excess above the escape velocity can only increase. The lower limit on the IMBH mass we infer from the velocities can also only increase when using the full 3D velocity.

R3-2

2. The authors emphasize that 3 of the innermost and fastest moving stars are also the faintest in the sample (Stars A, C, and D; lines 131 to 141). They use this to hint at a possible ejection/capture scenario for the existence of a population of low mass, very high velocity stars. However, the faint magnitudes of these stars also imply that they are the most likely to have large uncertainties in their astrometry and photometry. There is considerable discussion in the paper about the astrometric uncertainties, but it’s not clear how robustly these stars can be associated with the main sequence of wCen, which also calls their cluster membership into question. The authors need to discuss this in more detail since cluster membership is fundamental to all of their analysis.

We thank the referee for this suggestion. We added a discussion about the photometric errors in the Methods section.

Indeed we have to use a wider selection at fainter magnitudes for our CMD membership cuts to account for increased photometric errors, which leads to an apparent broadening of the main-sequence (see Figure 2). This looser cut can add additional foreground/background Milky Way contaminants to the sample.

However, we already account for this, by comparing the density of fast moving stars both with theoretical Milky Way models (and apply the same CMD selections) and by measuring the fast-star density at larger radii within our own dataset. While 1 contaminant among the fast stars is certainly possible ($p \sim 0.07$), more than 2 contaminants are strongly ruled out just from a statistical point of view.

Also, if the fast stars in the center would only enter the cluster sample due to photometric errors, we would expect a similar fraction and density also at larger radii. This is not the case (see Extended Data Figure 4 and 6).

R3-3

3. The discussion of high velocities associated with ejection from triple systems is insufficient (lines 177-185). The authors must be more specific about how many additional high velocity stars would be expected at larger radii if ejections are common. The authors also claim the

“high rate” of fast moving stars implies that the cluster would be depleted rapidly if ejection were common, but they do not clarify what rate they are using, e.g., is it based on the 5-star sample or the 7-star sample? What if all but 2 of these stars are contaminants? This scenario requires more detailed treatment since dynamical interactions are well-known to occur in massive stellar clusters like wCen.

Detailed explanations about these scenarios have been moved to the Methods section. We changed the wording of the discussion and added a few references to the literature about ejection mechanisms and expected rates. In addition we added the details of the numerical calculations for the estimated ejection rate.

R3-4

4. In lines 197 to 210, the authors mention that other MWL constraints taken together with their results imply an extremely low Eddington ratio for their claimed IMBH. They even point out that newly discovered pulsars in wCen might offer a direct measure of the gas density in the cluster, but they do not pursue this possibility. The authors should make at least a rough estimate of the gas density based on the pulsar DMs (if they are published) and check whether it is consistent with the low Eddington ratio they require.

As R1 pointed out, the expected luminosity of the IMBH depends on the surrounding gas density, accretion rate and radiative efficiency, significantly complicating any conclusions we can draw even if we have the surrounding gas density. We have removed the text about accretion from the main body of the paper due to space constraints, while the section in the Methods section now just highlights the low accretion rate and discusses prospects for future detections. We defer a more thorough discussion of accretion constraints to a future paper.

Minor Comments

I have a series of other, more minor comments on the analysis, text, figures and tables:

R3-m1

1. In line 232 and lines 236-237, the authors call the PM catalog “high-precision” and then claim “unprecedented depth and precision”. These statements need to be quantified and ideally associated with a literature reference.

We quantified the proper motion precision of stars in the center of our catalog. A detailed discussion and comparison with other proper motion datasets is given in the Häberle et al. catalog paper, for which we added a reference.

In addition the depth of the different available HST catalogs was already compared in the section “Comparison with other proper motion datasets” and in Figure 6.

R3-m2

2. The acronym MGE is never defined (first used line 371), I assume this is “multi-Gaussian expansion” but the authors should clarify and explain (briefly) why these are the appropriate models to use.

Thank you! This whole section has been significantly extended and we also properly define “MGE”

R3-m3

3. Lines 435-436 — it's not clear what the "delta" is in reference to; is this relative to the Anderson'10 cluster centre?

Yes, the delta refers to the Anderson10/ AvdM10 center. We have slightly changed the wording to make this clear.

R3-m4

4. Figure 1, left panel: the magenta on blue in this figure is difficult to make out. Please consider making the blue symbols light grey to improve readability. The proper motion vectors in the right-hand panel are also hard to make out; perhaps the authors could remove the legend on the right-hand-side (since it is the same in both panels) and zoom in a bit further.

We have changed the color scheme of Figure 1 and adapted the legend to improve readability.

R3-m5

5. Figure 3 — no error bars are given for the colour-magnitude diagram so it's not clear how confidently associated the fast-moving stars are with the cluster main-sequence, particularly at the faint end; perhaps a "characteristic" set of error bars for the brightest and faint target could be added for reference.

We have added error bars to the CMD and also list the statistical photometric errors in Table 3 & 4.

R3-m6

6. Similar comment for Fig 5 — it's not clear how close or far the targets of interest are from the primary sequences of the cluster stars since there are no error bars.

The quantities shown in Figure 5 are diagnostic parameters for the quality of the photometry and therefore do not have an intrinsic error and do not discriminate between cluster/foreground stars.

However, we now added two tables (Table 3 & 4) in which we report the numerical values and the percentile of the fast stars' quality parameters in comparison with stars at similar magnitudes. All stars in our robust sample have typical values for stars of their magnitude.

R3-m7

7. In Figure 6, Stars B, E, and G appear to suffer from confusion. Stars B and G are often omitted in the analysis, but Star E is not similarly flagged. Why keep Stars B and G in the analysis at all? They muddy the narrative and even the possible acceleration measurement for Star B does not improve the fit. I would like to see a concrete argument for including Stars B and G, otherwise I suggest removing them from the sample.

Star B and G fulfill all astrometric quality criteria and have an inferred velocity higher than the escape velocity. Therefore, we think it is necessary to report them.

However, as their velocity excess is not significant (1.1 sigma and 2.2 sigma) and indeed we have some concerns about the validity of their measurement due to crowding and low quality

of the PSF fit, we excluded them from the sample we consider robust. Including them would not change our conclusions and our numerical results (as the strongest constraints are provided by the fastest stars).

Indeed, also star E shows a close neighboring source, however, it is significantly brighter than stars B/G and therefore we are less concerned about the quality of the measurements.

R3-m8

8. Figure 7 — the two blue histograms are very similar and hard to tell apart. Also, it looks like the distribution in magnitude is bimodal. If all of the fast-moving stars are in one part of the bimodal distribution, might this impact their claimed association with the CMD and thus their cluster membership?

We have changed the color and style of the histograms to improve their readability and also added the magnitude distribution of available Gaia measurements.

The bimodal distribution of the histogram is a true physical effect and a consequence of the distribution of stellar masses in wCen and their luminosity function. Indeed it is interesting (though not strongly significant) that the 4 innermost fast moving stars are all faint and therefore in the right part of the distribution, we already discuss this in the manuscript.

In our analysis of the expected Milky Way contamination we take into account the CMD position of the Milky Way stars and find only a very low density of contaminating stars in the allowed CMD region (also at faint magnitudes), therefore we do not think there is reason to be concerned about the specific magnitude distribution of the fast moving stars in terms of cluster membership.

R3-m9

9. There are two typos in the caption for Figure 9: In the second sentence, it should be “profiles” and in the 3rd-to-last line it should be “pink markers”.

We fixed the typos. Thank you!

Reviewer Reports on the First Revision:

Referees' comments:

Referee #1 (Remarks to the Author):

In my initial report I had no fundamental reservations about the central, exciting results of the paper. I find the revised version to be compelling, with again no significant flaws, but improved in the presentation of the results and their limitations. If anything, the revised version is perhaps a touch conservative, but the main results and their supporting information are clear.

I recommend the paper for acceptance to Nature, and look forward to seeing the follow-up observations that can further constrain the location and mass of the black hole.

Referee #2 (Remarks to the Author):

I thank the authors for their responses to my concerns, and to those of the other referees. I am very happy to see this work published.

Referee #3 (Remarks to the Author):

Conclusive evidence for an intermediate-mass black hole in ω Centauri
by Häberle and co-authors
Nature manuscript 2023-12-22619

These results remain compelling and the presentation of the work is improved in this revision. I am particularly glad to see the more robust section on photometric errors and an expanded discussion of the ejection scenarios, both in the Methods section, though I hope more work will be done on 3- and 4-body interactions in the future work. While I respect the authors' decision to remove any mention of the pulsars in ω Cen, I hope these too will be returned to as these may offer probes of the gas density in the cluster and/or offer a constraint on dynamical interactions.

With the implementation of few minor suggestions listed below, I support publication of this work in Nature.

1. The authors are still attracted to overstated language, e.g., in the last sentence they claim "These results confirm ω Centauri hosts an IMBH and the nearest known massive black hole." -- instead of "confirm" I would favour "argue for" or "support". As is repeated by more than one reviewer, the result can and should stand on its own, without needing overstatement. I appreciate that the authors have mostly toned this down throughout the rest of the revision.

2. Figure 2 — I appreciate the photometric error bars on the pink points, but find the listing of the velocities for each fast-moving star distracting (particularly since these also appear in the Table in the Extended Data section). I suggest those be removed and that the letter labels appear in a larger font.

4. A few small grammatical things, which may be personal style and/or can be ironed out with copy editors down the line:

line 95: “improbably” → “unlikely”

line 98: “implications on” → “implications for”

line 135: Not sure what “(Moved this sentence:)” is referring to — the sentence that is still there or one that was there previously? Maybe this refers to line 227?

line 136: “signal by” → “signal from”

line 153: → “*the* fraction”

line 182: “also motivates to revisit” → “also motivates *us* to revisit” (if that’s the intended meaning?)

line 351: not sure what the word “similar” is doing at the end of this sentence.

line 864: I do not understand why there are tables in “panels” (d) and (e); these should be separated from the figure. (Similar at line 937.)

In general, this version of the manuscript would benefit from a thorough grammar check — the new text in the revision reads as if it was somewhat rushed, rather than carefully composed and/or iterated on by the co-authors.

Author Rebuttals to First Revision:

Referee #1 (Remarks to the Author):

In my initial report I had no fundamental reservations about the central, exciting results of the paper. I find the revised version to be compelling, with again no significant flaws, but improved in the presentation of the results and their limitations. If anything, the revised version is perhaps a touch conservative, but the main results and their supporting information are clear.

I recommend the paper for acceptance to Nature, and look forward to seeing the follow-up observations that can further constrain the location and mass of the black hole.

Referee #2 (Remarks to the Author):

I thank the authors for their responses to my concerns, and to those of the other referees. I am very happy to see this work published.

Referee #3 (Remarks to the Author):

Conclusive evidence for an intermediate-mass black hole in ω Centauri
by Häberle and co-authors
Nature manuscript 2023-12-22619

These results remain compelling and the presentation of the work is improved in this revision. I am particularly glad to see the more robust section on photometric errors and an expanded discussion of the ejection scenarios, both in the Methods section, though I hope more work will be done on 3- and 4-body interactions in the future work. While I respect the authors' decision to remove any mention of the pulsars in ω Cen, I hope these too will be returned to as these may offer probes of the gas density in the cluster and/or offer a constraint on dynamical interactions.

With the implementation of few minor suggestions listed below, I support publication of this work in Nature.

1. The authors are still attracted to overstated language, e.g., in the last sentence they claim "These results confirm ω Centauri hosts an IMBH and the nearest known massive black hole." -- instead of "confirm" I would favour "argue for" or "support". As is repeated by more than one reviewer, the result can and should stand on its own, without needing overstatement. I appreciate that the authors have mostly toned this down throughout the rest of the revision.

We have removed this last sentence.

2. Figure 2 — I appreciate the photometric error bars on the pink points, but find the listing of the velocities for each fast-moving star distracting (particularly since these also appear in the

Table in the Extended Data section). I suggest those be removed and that the letter labels appear in a larger font.

We have removed the velocities from Figure 2 and have added them to Figure 1 (see above)

4. A few small grammatical things, which may be personal style and/or can be ironed out with copy editors down the line:

line 95: “improbably” → “unlikely”

We have implemented the suggested change.

line 98: “implications on” → “implications for”

We have implemented the suggested change.

line 135: Not sure what “(Moved this sentence:)” is referring to — the sentence that is still there or one that was there previously? Maybe this refers to line 227?

In the initial submission, the discussion about linear motions and accelerations was mentioned at an earlier point in the manuscript, we have moved it to its current position.

line 136: “signal by” → “signal from”

We have implemented the suggested change.

line 153: → “*the* fraction”

We have implemented the suggested change.

line 182: “also motivates to revisit” → “also motivates *us* to revisit” (if that’s the intended meaning?)

We have reworded this sentence to fix it grammatically and clarify its meaning.

line 351: not sure what the word “similar” is doing at the end of this sentence.

We have removed this.

line 864: I do not understand why there are tables in “panels” (d) and (e); these should be separated from the figure. (Similar at line 937.)

We have restructured the Extended Data part, there are no more combined Figures / Tables.

In general, this version of the manuscript would benefit from a thorough grammar check — the new text in the revision reads as if it was somewhat rushed, rather than carefully composed and/or iterated on by the co-authors.